# Learning Equivariant Models by Discovering Symmetries with Learnable Augmentations

## Abstract

Recently, a trend has emerged that favors shifting away from designing constrained equivariant architectures for data in geometric domains and instead (1) modifying the training protocol, e.g., with a specific loss and data augmentations (soft equivariance), or (2) ignoring equivariance and inferring it only implicitly. However, both options have limitations, e.g., soft equivariance still requires a priori knowledge about the underlying symmetries, while implicitly learning equivariance from data lacks interpretability. To address these limitations, we propose SEMoLA, an end-to-end approach that jointly (1) discovers a priori unknown symmetries in the data via learnable data augmentations, and uses them to (2) encode the respective approximate equivariance into arbitrary unconstrained models. Hence, it enables learning equivariant models that do not need prior knowledge about symmetries, offer interpretability, and maintain robustness to distribution shifts. Empirically, we demonstrate the ability of SEMoLA to robustly discover relevant symmetries while achieving high prediction performance across various datasets, encompassing multiple data modalities and underlying symmetry groups.

## 1 Introduction

In recent years, equivariant machine learning models have consolidated themselves as a powerful approach for working with geometric data, boasting increased data efficiency and generalization capabilities (Dehmamy et al., 2021; Cohen and Welling, 2016; Bulusu et al., 2021). In particular, they have shown great promise in their application to scientific domains, e.g., dynamical systems (Han et al., 2022; Xu et al., 2024), chemistry (Satorras et al., 2021; Brandstetter et al., 2021), or structural biology (Jumper et al., 2021), where the strong inductive biases imposed by the equivariant design of the models can lead to improved performance and scientific validity.

However, recently, some high-profile works (Abramson et al., 2024; Wang et al., 2024) have begun to question the need of using equivariant models for certain tasks in light of the ability of high-capacity unconstrained models to achieve similar performance without the intrinsic limitations of architectures with hard-coded equivariance, raising the question of whether equivariance actually matters at scale (Brehmer et al., 2024). These limitations include implementation complexity, scalability issues, excessively constrained hypothesis space (for tasks that involve approximate equivariance due to noisy or imperfect data), or requiring a priori knowledge of the symmetry group.

Besides ignoring the equivariance and using high-capacity fully unconstrained models, the other alternative to 'hard' equivariant models is 'soft' equivariant models, i.e., unconstrained architectures for which the training protocol is adjusted to encourage equivariance. The most common such approach is data augmentation (Gerken et al., 2022; Quiroga et al., 2020; Xu et al., 2023), but recently, multi-task learning has also been proposed as a way of controlling the extent to which the base model is encouraged to be approximately equivariant (Elhag et al., 2024).

However, these two approaches come with their own limitations. On the one hand, 'soft' equivariant models still demand a priori additional knowledge, for instance, about the symmetry group, which often may not be readily available or require domain expertise. On the other hand, fully unconstrained models lack guarantees for learning any relevant symmetry, and they also lack explicit interpretability of their internal representations to allow ascertaining such behavior.

This raises the question of whether it is possible to learn symmetry-aware models but without direct supervision on the symmetry. A recent line of work has explored the problem of automatic symmetry discovery, i.e., explicitly discovering unknown symmetry groups to which a dataset is equivariant solely from the data (Yang et al., 2023; Benton et al., 2020; Yang et al., 2024; Desai et al., 2022). Nevertheless, so far, most of these approaches have been devised as a single step in a model design pipeline, where the symmetry discovery approach is decoupled from the downstream task and hence cannot adjust if only specific symmetries are relevant for the task. Moreover, unsupervised approaches may need any symmetry in the downstream task (e.g., rotation-invariant labels) to be a symmetry in the data (feature) distribution (e.g., the data needs to be uniformly distributed over rotation angles).

In this paper, we propose SEMoLA (Soft-Equivariant Models with Learnable Augmentations), an approach that seeks to overcome these limitations by allowing equivariance to an unknown symmetry to be automatically learned from data while producing predictions through an interpretable end-to-end method. The main contributions of our work are:

- We introduce an end-to-end approach that discovers a priori unknown continuous symmetries using learnable data augmentations, which are used to jointly encode approximate equivariance to the discovered symmetries in an arbitrary underlying unconstrained model.
- Our results show that this approach can improve the performance, sample efficiency, and robustness out-of-distribution of symmetry-agnostic models, and are comparable to those of 'hard' and 'soft' equivariant models designed with the ground truth symmetry.
- In addition, our approach learns equivariance robustly from both the data and task, while being interpretable by containing an explicit representation of the learned symmetry, and allowing control over how much the equivariance is enforced into the model.

## 2  RELATED WORK

**Equivariant machine learning.**  Many works have explored the design of equivariant models for groups known a priori, such as permutations in sets (Zaheer et al., 2017), translations (Krizhevsky et al., 2012), roto-translations (Fuchs et al., 2020), or local gauge transformations (Cohen et al., 2019). There are also more general approaches that can be adapted to different specified symmetry groups, such as EMLP (Finzi et al., 2021) and GCNN (Cohen and Welling, 2016). Instead of designing a constrained equivariant model (hard equivariance), other approaches opt for learning the symmetries (soft equivariance) from data through pre-specified augmentations (Hu et al., 2021; Gerken et al., 2022; Wang et al., 2022; Quiroga et al., 2020; Chen et al., 2020; Xu et al., 2023) or multi-task learning (Elhag et al., 2024). Recent works (Abramson et al., 2024; Wang et al., 2024) have shown that large-scale unconstrained architectures can be a valid alternative when having access to enough data. While this last approach is similar to our proposal in that it does not consider a pre-specified symmetry, it fails to provide any insight into what equivariance, if any, the models learn. Another related approach is the soft equivariant model REMUL (Elhag et al., 2024), which frames learning equivariance as a weighted multi-task problem for unconstrained models. However, while that method considers a known symmetry group, our proposal tackles the more complex scenario of discovering the correct symmetry while jointly encoding it into the unconstrained model. Another set of related works correspond to the unsupervised representation learning of latent spaces that disentangle the action of a potentially unknown group on the original input space to facilitate the application of simpler equivariant models that operate on that pretrained latent space (Koyama et al., 2023; Mitchel et al., 2024). These works, however, are not designed to produce interpretable representations of the possibly unknown group or to directly generate approximately equivariant predictions, and, thus, also differ from the problems that our method aims to address.

**Symmetry discovery.**  Recently, multiple methods for automatically discovering symmetries in datasets across different search spaces have been proposed, e.g., Zhou et al. (2021); Karjol et al. (2024) discover discrete finite groups, Augerino (Benton et al., 2020) learns subsets of specified groups, SymmetryGAN (Desai et al., 2022) learns individual symmetry transformations on the dataset distribution, LieGAN (Yang et al., 2023) discovers continuous symmetries, and LaLiGAN (Yang et al., 2024) learns non-linear continuous symmetries. These approaches were introduced as a first step in a model design pipeline, where, after training an initial model to discover the symmetry, this symmetry is used to define an equivariant model that produces predictions. Similarly, other works

explore the reversed scenario of defining models that identify symmetries from a previously trained prediction model (Krippendorf and Syvaeri, 2020; Moskalev et al., 2023; Hu et al., 2024). Other approaches attempt to discover symmetries through learnable weight-sharing schemes (der Linden et al., 2024), specifically designed architectures (Dehmamy et al., 2021), or approximate Bayesian model selection (van der Ouderaa et al., 2024). However, they suffer from additional complexity and assumptions, limited scalability, and reduced interpretability. For instance, L-conv (Dehmamy et al., 2021) suffers from high computational complexity due to the computation of exponential maps in every layer of their hard-constrained architecture, which is also not well-suited for dealing with approximate and inexact symmetries. Most related to our work are LieGAN (Yang et al., 2023) and Augerino (Benton et al., 2020), since they also discover continuous symmetries with Lie algebra bases in an architecture-agnostic manner. However, LieGAN (Yang et al., 2023) does not produce predictions and discovers symmetries as transformations that preserve the data distribution, which makes its performance sensitive to the relevant transformations being uniformly distributed in the data (Hu et al., 2024), while our approach is more robust due to the additional learning signal from the prediction task. Augerino (Benton et al., 2020) produces predictions but is restricted to discovering a distribution over coefficients associated with the generators of a given group rather than a fully unknown group, as we propose, and its training and inference differ from ours, e.g., it does not allow control over how much equivariance is enforced into the architecture, thus lacking adaptability to different datasets and scenarios. A modified version, Augerino+, was introduced in (Yang et al., 2023) to explore its ability to learn unknown symmetry groups, but the experimental results in that paper suggest that it is not applicable for this purpose.

**Learnable augmentations.**   An integral part of our proposal consists of defining a data augmentation process that, rather than relying on predefined transformations, as is predominantly common in the literature (Gerken et al., 2022; Quiroga et al., 2020; Xu et al., 2023), has learnable parameters that are trained jointly with the prediction model. Learnable augmentations have already been explored in the context, for instance, of anomaly detection for time series (Qiu et al., 2021), or self-supervised GNNs (Kefato et al., 2021), showing that they are capable of providing additional flexibility and enhanced performance. Nevertheless, in previous works, the learnable augmentations are usually represented by black-box models, such as Multi-Layer Perceptrons (MLPs), which hinders the interpretability of the generated augmentations. In contrast, our approach parameterizes the learnable augmentations as a Lie algebra basis, as is common in symmetry discovery methods (Yang et al., 2023; 2024; Benton et al., 2020), thus enabling a straightforward interpretation of the augmentations, which allows for greater insight into characteristics of the data that could be of interest to practitioners.

## 3 BACKGROUND

**Equivariant models.**   We assume that we work with a dataset $\mathcal{D} = \{(x_i, y_i)\}_{i=1}^{N}$ that lives in a geometric domain with a symmetry group $G$. Models $f : \mathcal{X} \to \mathcal{Y}$ from the input domain $\mathcal{X} \subseteq \mathbb{R}^n$ to the output domain $\mathcal{Y} \subseteq \mathbb{R}^m$ are equivariant to the group if the actions of $G$ on $\mathcal{X}$ correspond to related actions on $\mathcal{Y}$, through the appropriate group representations:

**Definition 3.1.** A function $f : \mathcal{X} \to \mathcal{Y}$ is equivariant with respect to a group $G$ if $\forall g \in G, (x, y) \in \mathcal{D}$ we have $f(\rho_{\mathcal{X}}(g)x) = \rho_{\mathcal{Y}}(g)f(x)$, where $\rho_{\mathcal{X}} : G \to \mathrm{GL}(n)$ and $\rho_{\mathcal{Y}} : G \to \mathrm{GL}(m)$ are representations of the group $G$ acting on the input $\mathcal{X}$ and output $\mathcal{Y}$, respectively. Invariance can be seen as a special case of equivariance, where $\rho_{\mathcal{Y}} = \mathrm{Id}$.

**Definition 3.2.** A group representation $\rho : G \to \mathrm{GL}(n)$ is a group homomorphism that maps every group element $g \in G$ to a non-singular matrix $\rho(g) \in \mathbb{R}^{n \times n}$ that can be applied to transform the data from $\mathcal{X}$. We will sometimes omit $\rho_{\mathcal{X}}$ and $\rho_{\mathcal{Y}}$ when clear and simply write $gx$ or $gy$ instead.

**Lie groups.**   Our approach focuses on learning continuous symmetries, which are most naturally modeled by Lie groups, i.e., groups that are also differentiable manifolds, due to their relevance for many real-world tasks, especially in scientific domains. Some examples include rotation groups $SO(n)$, or Euclidean groups $E(n)$. Many relevant Lie groups are matrix Lie groups, since they can be expressed as subgroups of the Lie group $\mathrm{GL}(n, \mathbb{R})$, and their group product is the matrix product. Each Lie group $G$ is associated with a Lie algebra, which is the tangent space at the identity $\mathfrak{g} = T_{\mathrm{Id}}G$, and is a vector space of the same dimensionality as the group. A basis of the Lie algebra $L_i \in \mathfrak{g}$ can be viewed as the set of infinitesimal generators of the associated Lie group $G$. To map Lie algebra

elements to elements of the Lie group, we can apply the exponential map $\exp : \mathfrak{g} \to G$, which, for instance, for matrix groups corresponds to the matrix exponential, i.e., $\exp(A) = \sum_{n=0}^{\infty} \frac{1}{n!} A^n$. If the Lie group $G$ is connected and compact, we can write its elements as $g = \exp(\sum_i w_i L_i)$, with this map being surjective. This parameterization using Lie algebras, which coincides with that of other existing continuous symmetry discovery methods (e.g., LieGAN (Yang et al., 2023) and Augerino (Benton et al., 2020)), therefore has the limitation that it can only represent a single connected component of the associated Lie groups. For non-connected groups, the description of all their group elements would require a more complex parameterization, which, like other existing symmetry discovery methods, we do not consider in this work.

## 4  METHOD

In this section, we present our approach, SEMoLA (Soft-Equivariant Models with Learnable Augmentations), illustrated in Figure 1. Its main components are the learnable LieAugmenter module that generates augmentations from the original input data using the discovered symmetry, and an arbitrary unconstrained base model that generates the predictions. In the following, we provide a more detailed description of these components as well as the training and inference processes.

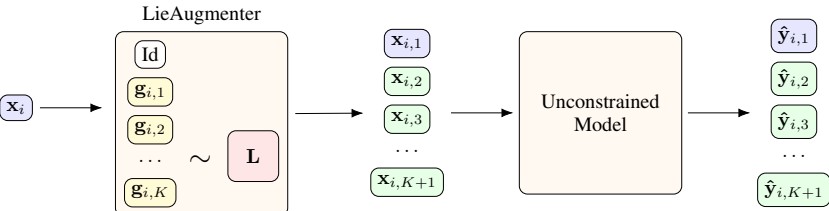

Figure 1: Structure of SEMoLA. The LieAugmenter module learns a Lie algebra basis, from which Lie group elements are sampled to augment the original input data. An arbitrary unconstrained base model then processes both the original and augmented data and outputs the corresponding predictions.

### 4.1  LIEAUGMENTER

Our learnable augmenter module models continuous symmetries using Lie theory, following an approach similar to some symmetry discovery methods (Benton et al., 2020; Yang et al., 2023), which, while limited to connected Lie groups, is applicable to a wide range of relevant and common symmetry groups (e.g., $SO(n)$, $SE(n)$). Moreover, through alternative parameterization strategies, it can also be applied to other types of symmetries, such as discrete subgroups, as shown in Appendix A.8.

In particular, we reparameterize a distribution over the elements of a matrix Lie group $G$ through its generators, which are given by a Lie algebra basis with cardinality $C$ and dimensionality $D$, $L = \{L_i \in \mathbb{R}^{D \times D}\}_{i=1}^C$, whose entries we define as parameters that are optimized during training. The value of $D$ is known from the dimensionality of the data, but $C$ needs to be estimated in general. In our experiments, we assume that the value of $C$ for the correct symmetry group is known, but in Appendix A.9 we also show that our approach can be robust to misspecifications.

We can then sample elements from the Lie group by sampling coefficients from a distribution $w \sim P(\gamma)$, and then applying the matrix exponential to the resulting linear combination of elements of the Lie algebra basis $g = \exp\left[\sum_{i=1}^C w_i L_i\right]$ (Falorsi et al., 2019). Unless otherwise stated, in our experiments, we sample the weights from a uniform distribution $P(\gamma) = \mathcal{U}[-\gamma, \gamma]$ to avoid any unwanted bias towards sampling a specific subset of group elements.

The LieAugmenter samples $K$ group elements per input sample, and computes the augmentations as the product of each sampled group element $g \in G$ and the corresponding input sample $x \in \mathcal{X}$. We explore the impact of the chosen number of augmentations on the performance of the approach in Appendix A.9, showing that our method is robust to that choice. We initialize all the parameters of the basis to a constant small value, e.g., 1e-2, to provide an uninformative prior on its potential structure, so that the learning process is not influenced by potentially beneficial or adversarial random

initializations. In addition, we normalize the parameters as $L_i' = \frac{\sqrt{D}}{||L_i||} L_i$, with $||L_i||$ being the Frobenius norm. This way, the basis maintains a constant norm throughout training for greater stability. The scale factor depends on the dimensionality of the basis, and thus encourages a more consistent representation across groups of different sizes.

## 4.2 BASE UNCONSTRAINED MODEL

After sampling augmentations of the input data from the LieAugmenter, these are fed into a base model $f_\theta$, which, as we previously mentioned, can be any unconstrained architecture as long as it is compatible with the considered data and task. For instance, in our main experiments, we use a CNN model when working with image data and an MLP for tensor inputs. Nevertheless, there is no a priori restriction on substituting such models for other more complex alternatives, and we explore some of them, e.g., ViTs (Dosovitskiy et al., 2020) and EfficientNets (Tan and Le, 2019), in Appendix A.7. In fact, a model that provides good performance for the task may lead to a faster convergence of the Lie algebra in the LieAugmenter to the correct symmetry, since the loss would be minimized more easily.

## 4.3 TRAINING

We formulate the training loss as a weighted multi-task objective that balances multiple considerations:

$$
\begin{aligned}
\mathcal{L}_{total}(f_\theta, \mathcal{X}, \mathcal{Y}, G, L) = {} & \alpha \mathcal{L}_{obj}(f_\theta, \mathcal{X}, \mathcal{Y}) + \beta \mathcal{L}_{equiv}(f_\theta, \mathcal{X}, \mathcal{Y}, G) \\
& + \lambda\, l_{areg}(\mathcal{X}, G) + \eta\, l_{bcreg}(L) + \nu\, l_{bsreg}(L)
\end{aligned}
\tag{1}
$$

The two main loss terms are the empirical loss $\mathcal{L}_{obj}$ (e.g., MSE, binary cross-entropy) and the equivariance loss $\mathcal{L}_{equiv}(f_\theta, \mathcal{X}, \mathcal{Y}, G) = \frac{1}{N} \sum_{i=1}^{N} \frac{1}{K} \sum_{j=1}^{K} \ell(f_\theta(\rho_\mathcal{X}(g_{i,j})(x_i)), \rho_\mathcal{Y}(g_{i,j})(y_i))$, with $\ell$ being a metric, such as the $L_1$ norm, and $\rho_\mathcal{Y}(g_{i,j})$ being simply $g_{i,j}$ for equivariant tasks and the identity for invariant tasks. In addition, we include some regularization terms that focus on encouraging desirable properties of the Lie algebra basis learned by the LieAugmenter module: $l_{areg}(\mathcal{X}, G) = \frac{1}{N} \sum_{i=1}^{N} \sum_{j=1}^{K} \mathrm{CosSim}(x_i, g_{i,j} x_i)$ encourages non-trivial augmentations, $l_{bcreg}(L) = \sum_{i=1}^{C} \sum_{j=i+1}^{C} \mathrm{CosSim}(L_i, L_j)$ orthogonality of the elements in the basis, and $l_{bsreg}(L) = \sum_{i=1}^{C} |L_i|$ sparse basis representations. We generally set $\alpha, \beta > \lambda, \eta > \nu > 0$ according to the importance of each term, and we explore ablations of the values for those coefficients in Appendix A.9, which show that our method is very robust to the choice of coefficient values.

The intuition behind why this loss function may allow the LieAugmenter to learn the correct symmetry relies on the fact that its minimization is facilitated when the learned symmetry is compatible with that of the dataset, since otherwise minimizing $\mathcal{L}_{equiv}$ leads to a restriction of the hypothesis space that can negatively affect the ability of the model to minimize $\mathcal{L}_{obj}$.

## 4.4 INFERENCE

At inference time, there are two main potential strategies for generating predictions. The first, more robust strategy averages the inversely transformed outputs of the model for the augmented inputs, i.e.,

$$
\hat{y}_i = \frac{1}{K+1} \Big( \hat{y}_{i,1} + \sum_{j=1}^{K} \rho_\mathcal{Y}(g_{i,j}^{-1}) \hat{y}_{i,j+1} \Big),
\tag{2}
$$

where again $\rho_\mathcal{Y}(g_{i,j}^{-1})$ is equal to $g_{i,j}^{-1}$ for equivariant tasks and the identity for invariant ones. This group action is well-defined following the existence of a well-defined action on the ground truth labels and the training objective of our method, which mainly through the equivariance loss term, encourages the model to generate outputs that are consistent with the learned group actions.

The second strategy does not generate any augmentations at inference time and simply processes the original inputs directly with the base model, which might be preferable in scenarios where inference speed is paramount. In our experiments, we mainly apply the averaging strategy in Equation 2, but we also provide some results with the second strategy in Appendix A.4.3, which help confirm the expected trade-off of faster inference speed but slightly worse performance.

## 5 EXPERIMENTS

In our experiments, we probe how well SEMoLA recovers underlying symmetries, how equivariant the prediction model becomes, and its prediction performance on the target task, both in- and out-of-distribution. We test different underlying symmetries, prediction tasks, and data types. For further details about the experimental setting as well as additional results, please refer to Appendix A. In addition, to facilitate the reproduction of our experiments, we make our implementation available[1].

**Metrics.** To evaluate the symmetry discovery results of the different approaches, we mainly rely on a qualitative comparison of visualizations of the learned Lie algebra bases, but we also report approximate quantitative measurements in the form of primarily the absolute cosine similarity and also the Mean Absolute Error (MAE) with the ground truth. These quantitative metrics are not always fully reliable because they ignore alternative representations of the Lie algebra basis of the correct symmetry, but they still provide some insight into the robustness of the approaches. Nevertheless, we complement them by also considering the effect that applying augmentations from the learned symmetries has on the base model's performance compared to using the ground truth symmetries, which we believe provides a strong indication of the correctness of the discovered symmetries.

To quantify the equivariance learned by the models, similarly to Elhag et al. (2024), we apply the following equivariant error metric:

$$E(f_\theta, G) = \frac{1}{N} \sum_{i=1}^{N} \left\| \frac{1}{K} \sum_{j=1}^{K} \rho_{\mathcal{Y}}(g_j)(f_\theta(x_i)) - \frac{1}{K} \sum_{j=1}^{K} f_\theta(\rho_{\mathcal{X}}(g_j)(x_i)) \right\| \tag{3}$$

where $\|\|$ is defined to be the $L_1$ norm, and we consider $K = 10$.

**Baselines.** When analyzing the performance of our method, it is crucial to compare it against relevant baselines from the literature. However, to the best of our knowledge, there is no existing alternative that successfully performs interpretable discovery of continuous symmetries and equivariance learning while producing predictions with an unconstrained model, as we do. Thus, when defining the baselines, we can only consider approaches designed to address a subset of those tasks.

For the symmetry discovery task, we provide a comparison with LieGAN and Augerino+ (Yang et al., 2023). Then, for equivariance and prediction performance, we consider two 'hard' equivariant models that are adaptable to different symmetry groups, GCNN (Cohen and Welling, 2016) and EMLP (Finzi et al., 2021). We also provide the performance of the unconstrained model that we use for SEMoLA without the augmenter and also with a non-learnable augmenter that uses the ground truth Lie algebra basis, to establish approximate lower and upper bounds on the performance and equivariance achievable by our proposal. Lastly, we also report the performance of incorporating the Lie algebra basis discovered by LieGAN into both EMLP and the non-learnable augmenter, seeking to demonstrate the performance that the two-step process of discovering and then encoding a symmetry can accomplish.

### 5.1 ROTATEDMNIST

Firstly, we consider a version of the MNIST dataset (Deng, 2012) where we randomly rotate each image, thus establishing the rotation group SO(2) as the underlying symmetry we want to discover. In particular, we define two scenarios: in-distribution (in which the images of the train and test sets are randomly rotated by an angle in $[0, 360)$ degrees), and out-of-distribution (in which the images of the training set are randomly rotated by an angle in $[0, 90) \cup [0, -90)$ degrees and for the test set the rotation is by an angle in $[90, 180) \cup [-90, -180)$ degrees). This out-of-distribution setting mainly affects the symmetry discovery methods in that they have to learn the correct symmetry from a restricted and less diversely augmented training set, and the predictive models in that they need to be robust to differently distributed data between training and testing.

For the symmetry discovery methods, including SEMoLA's LieAugmenter module, we set the cardinality of the Lie algebra basis to $C = 1$ and search over the six-dimensional space of 2D affine transformations, which are defined to act on the pixel coordinates of the images. Even though

---

[1]https://anonymous.4open.science/r/SEMoLA/

this assumption introduces some prior knowledge into this particular symmetry discovery task, we believe that it still leads to meaningful results since it corresponds to common practical scenarios. As we can observe in Figure 2, SEMoLA and LieGAN can identify the correct symmetry for this dataset in both the in-distribution and out-of-distribution scenarios up to a sign (which does not impact the correctness since the sampling distribution $P(\gamma)$ is symmetric around 0), while Augerino+ performs worse in the in-distribution case, and fails to learn the correct Lie algebra basis in the out-of-distribution scenario. Moreover, from Table 1, we can see that while SEMoLA robustly discovers the correct symmetry in both the in-distribution and out-of-distribution cases, LieGAN is only robust in-distribution, i.e., it is sensitive to the non-uniform distribution of rotations in the out-of-distribution data, which is likely due to its objective for symmetry discovery that focuses on transformations that preserve the data distribution.

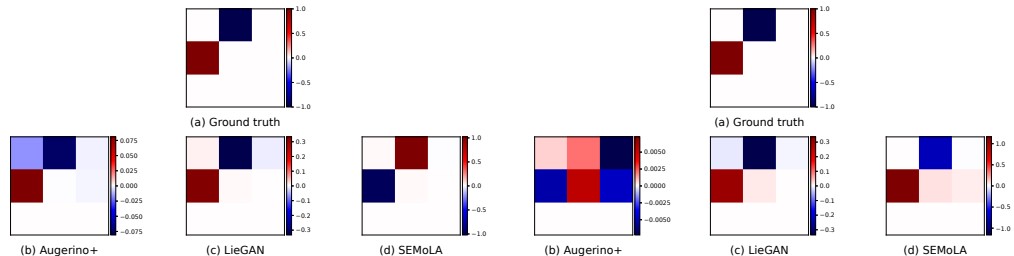

Figure 2: Comparison of the ground truth Lie algebra basis with those learned by LieGAN, Augerino+, and our proposed SEMoLA approach for the in-distribution version of the RotatedMNIST dataset (left) and the restricted out-of-distribution version (right).

Table 1: Cosine and MAE similarities of the ground truth Lie algebra basis and the ones learned by each of the considered symmetry learning models in the in-distribution and out-of-distribution versions of the RotatedMNIST dataset. For each metric, we report the mean and standard deviation across three runs using different random seeds.

| Model | In distribution | | Out of distribution | |
|---|---|---|---|---|
| | Cosine | MAE | Cosine | MAE |
| Augerino+ | $0.7883_{\pm 0.1407}$ | $0.2169_{\pm 0.0068}$ | $0.3418_{\pm 0.2208}$ | $0.2246_{\pm 0.0004}$ |
| LieGAN | $0.9998_{\pm 0.0002}$ | $0.1379_{\pm 0.0104}$ | $0.7490_{\pm 0.3540}$ | $0.1705_{\pm 0.0379}$ |
| SEMoLA | $0.9997_{\pm 0.0002}$ | $0.3077_{\pm 0.1989}$ | $0.9932_{\pm 0.0063}$ | $0.1940_{\pm 0.1956}$ |

In terms of classification performance, Table 2 shows that SEMoLA provides the overall second-best performance, only behind (but close to) the performance of the CNN model that considers ground truth augmentations. The CNN model with augmentations from the Lie algebra basis discovered by LieGAN also shows a similarly high performance, while the other baselines, i.e., Augerino+, CNN, and GCNN, all provide considerably lower accuracy in both scenarios. The performance of GCNN is perhaps unexpectedly low for a hard equivariant model, but the reason probably lies in its design for invariance to the $p4$ group, which does not exactly match the continuous symmetry of this dataset.

We provide further results for this dataset in Appendix A.4, such as analyzing the sample efficiency of the prediction approaches, from which we can see that our proposal mostly matches the performance of the base model with ground truth augmentations even with decreasing sizes of the training set.

## 5.2 N-BODY DYNAMICS

As the next experiment, we consider a simulated N-body trajectory dataset from Greydanus et al. (2019), generated as in Yang et al. (2023). It consists of two bodies with identical masses rotating in nearly circular orbits, and the objective is to predict future coordinates based on past observations, which is a rotationally equivariant problem. In general, the input and output feature vectors for each timestep have $4N$ entries, corresponding to the positions and momenta of the bodies: $[q_{1x}, q_{1y}, p_{1x}, p_{1y}, \ldots, q_{Nx}, q_{Ny}, p_{Nx}, p_{Ny}]$. Again, we define two scenarios: in-distribution (i.e.,

Table 2: Results for the classification performance and equivariant error of the considered models for in-distribution and out-of-distribution versions of the RotatedMNIST dataset. For each metric, we report the mean and standard deviation across three runs using different random seeds. The accuracies are expressed as percentages.

| | In distribution | | Out of distribution | |
|---|---|---|---|---|
| Model | Accuracy | Equiv. Error | Accuracy | Equiv. Error |
| Augerino+ | $96.85_{\pm 0.30}$ | $5.09_{\pm 0.70}$ | $57.68_{\pm 1.45}$ | $3.97_{\pm 0.51}$ |
| CNN | $96.17_{\pm 0.34}$ | $5.52_{\pm 0.23}$ | $55.24_{\pm 1.14}$ | $7.19_{\pm 0.54}$ |
| CNN + GT aug | $99.06_{\pm 0.00}$ | $4.01_{\pm 0.29}$ | $99.09_{\pm 0.08}$ | $3.87_{\pm 0.35}$ |
| CNN + LieGAN aug | $98.69_{\pm 0.16}$ | $5.04_{\pm 0.19}$ | $91.59_{\pm 1.07}$ | $5.82_{\pm 0.47}$ |
| GCNN | $96.78_{\pm 0.36}$ | $5.31_{\pm 1.16}$ | $55.60_{\pm 1.45}$ | $5.78_{\pm 1.23}$ |
| SEMoLA | $99.08_{\pm 0.06}$ | $3.10_{\pm 0.32}$ | $90.82_{\pm 6.49}$ | $3.26_{\pm 0.62}$ |

the default dataset), and out-of-distribution (in which the training set is modified to ensure that the first body's position is always in the top-left or bottom-right quadrants, and for the test set it is always in the top-right or bottom-left quadrants). Here, we focus our analysis on the task of predicting one output timestep from one input timestep, but we provide further results in Appendix A.5, including for the more challenging scenario of considering three input and output timesteps, which show that our approach performs similarly well in that setting.

As we can see in Figure 3, both LieGAN and Augerino+ fail to learn the correct symmetry. Conversely, SEMoLA discovers an alternative representation of the Lie algebra basis of the correct symmetry under the constraints of the considered dataset (see Appendix A.5.5). LieGAN is capable of identifying the correct symmetry but only in the in-distribution case and when complemented with auxiliary knowledge in the form of different masks that restrict the interaction between different features of the input (see Appendix A.5), while Augerino+ still fails.

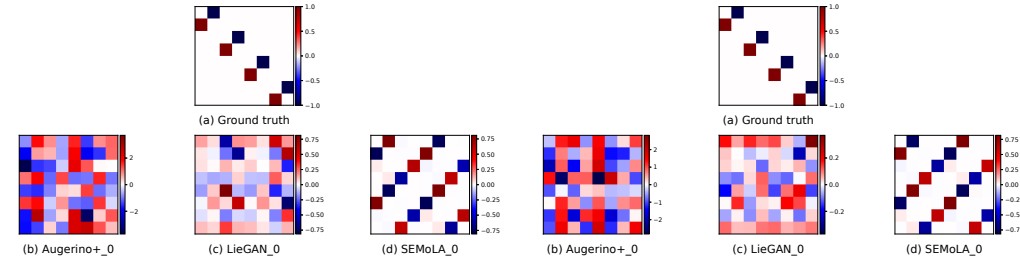

Figure 3: Comparison of the ground truth Lie algebra basis with those learned by LieGAN, Augerino+, and our proposed SEMoLA approach for the in-distribution version of the 2-body dataset (left) and the restricted out-of-distribution version (right) with one input and output timestep.

From the results in Table 3, we can see that SEMoLA provides the highest performance behind the hard-equivariant baselines that consider pre-specified symmetries, while outperforming Augerino+ and the base MLP (even when it considers augmentations from pre-specified symmetries) in terms of both task performance and equivariant error across in-distribution and out-of-distribution scenarios. We should note that the fact that SEMoLA performs slightly better than the MLP with ground truth augmentations is probably due to the alternative representation of the symmetry that it learns, which encourages more diverse rotations, as shown in Appendix A.5.5.

## 5.3 QM9

We also explore the performance of SEMoLA on the more complex task of molecular property prediction using the QM9 dataset (Blum and Reymond, 2009; Rupp et al., 2012), which consists of small inorganic molecules described by their 3D coordinates and atomic charges. We focus

Table 3: Results for the performance and equivariant errors of the considered models for in-distribution and out-of-distribution versions of the 2-body dataset with one input and output timestep. For each metric, we report the mean and standard deviation across three runs using different random seeds.

| Model | In distribution | | Out of distribution | |
| --- | --- | --- | --- | --- |
| | MSE | Equiv. Error | MSE | Equiv. Error |
| Augerino+ | 5.28e+00$_{\pm4.23e-01}$ | 1.90e-01$_{\pm1.92e-03}$ | 3.26e+00$_{\pm7.30e-01}$ | 1.94e-01$_{\pm4.15e-03}$ |
| MLP | 5.16e-03$_{\pm2.79e-03}$ | 2.48e-02$_{\pm7.37e-03}$ | 1.86e-02$_{\pm3.74e-03}$ | 3.33e-02$_{\pm2.56e-03}$ |
| MLP + GT aug | 2.13e-05$_{\pm1.56e-05}$ | 3.03e-03$_{\pm1.03e-03}$ | 3.03e-05$_{\pm8.81e-06}$ | 3.70e-03$_{\pm1.14e-03}$ |
| MLP + LieGAN aug | 3.23e-03$_{\pm6.25e-04}$ | 1.91e-02$_{\pm1.47e-03}$ | 1.15e-02$_{\pm3.00e-03}$ | 2.84e-02$_{\pm3.04e-03}$ |
| EMLP + GT | 1.74e-06$_{\pm1.50e-06}$ | 5.32e-05$_{\pm1.16e-05}$ | 4.64e-06$_{\pm4.86e-06}$ | 5.27e-05$_{\pm1.10e-05}$ |
| EMLP + LieGAN | 6.36e-06$_{\pm4.56e-06}$ | 5.31e-05$_{\pm1.13e-05}$ | 3.81e-06$_{\pm2.62e-06}$ | 5.31e-05$_{\pm1.09e-05}$ |
| SEMoLA | 1.39e-05$_{\pm6.08e-07}$ | 2.78e-03$_{\pm4.88e-04}$ | 2.38e-05$_{\pm9.04e-06}$ | 2.59e-03$_{\pm9.89e-05}$ |

on the tasks of predicting the highest occupied molecular orbital energy (HOMO) and the lowest unoccupied molecular orbital energy (LUMO), which are invariant under transformations of the atomic coordinates given by the SE(3) group. Thus, they provide an ideal scenario for testing the capability of our approach for learning more complex symmetries, consisting of multiple elements in their Lie algebra basis. We consider the LieConv model (Finzi et al., 2020) with no equivariance (LieConv-Trivial) as the base model for our SEMoLA approach, and compare its performance with that reported for different augmentation strategies, including using the ground truth and the symmetries discovered by LieGAN and Augerino (Benton et al., 2020).

As we can see in Figure 4, SEMoLA learns the correct Lie algebra basis of the SE(3) group up to permutation (which is correct due to the permutation-invariant nature of our strategy for sampling group elements) for both prediction tasks, and from the results in Table 4, we see that its performance is comparable to that of augmentations with the ground truth symmetry and that it outperforms using symmetries learned by LieGAN and Augerino.

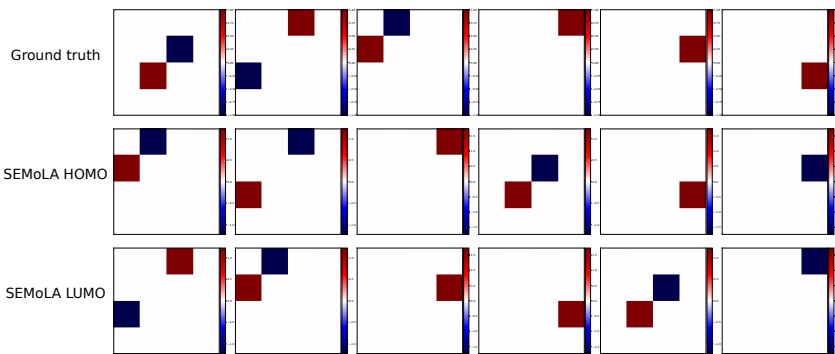

Figure 4: Comparison of the ground truth Lie algebra basis and those learned by SEMoLA for the HOMO and LUMO targets of the QM9 dataset.

## 6 CONCLUSION

In this paper, we present SEMoLA, an approach for defining learnable equivariant models without prior knowledge of the underlying symmetry, while providing interpretability, high-performant predictions, and low equivariance errors, as shown across multiple experiments. In particular, our results show that our method is able to match or surpass the performance of models specifically designed for either symmetry discovery or equivariant predictions, while jointly tackling both tasks.

While hard equivariant models still have many advantages (e.g., exact equivariance guarantees, higher sample and parameter efficiency, etc.) in cases where the symmetry is known a priori and exact, their adoption is limited by the complexity of their design and implementation. In this context, our

Table 4: Results for the performance and equivariant errors in meVs of the different considered models for the QM9 dataset. The results for the models marked with [†] are from Benton et al. (2020) and those marked with [‡] come from Yang et al. (2023).

| | HOMO | | LUMO | |
|---|---|---|---|---|
| Model | MAE | Equiv. Error | MAE | Equiv. Error |
| No Symmetry[†] | 52.7 | - | 43.5 | - |
| SE(3)[†] | 36.5 | - | 29.8 | - |
| Augerino[†] | 38.3 | - | 33.7 | - |
| LieGAN[‡] | 43.5 | - | 36.4 | - |
| SEMoLA | 37.09 | 0.0033 | 30.67 | 0.0015 |

work can be seen as an initial attempt at streamlining the design, flexibility, interpretability, and applicability of equivariant models, which we hope can lead to a more widespread adoption of these models, especially when the underlying symmetries are inexact or harder to ascertain, as is quite common for real-world tasks in scientific domains, such as physics or chemistry.

Nevertheless, there are still many potential research directions based on our approach that could be explored in future work. For instance, modeling learnable non-connected Lie groups and non-linear symmetries, or applying our method to tasks without clearly known symmetries.

## REPRODUCIBILITY STATEMENT

We believe that our results being reproducible is of great importance to the correctness and usefulness of our work, so we make an effort to provide as many details as possible to ensure their reproducibility. In that sense, our description of the proposed method in Section 4 and of the experiments and results in Section 5, together with the additional details presented throughout Appendix A, covers the most relevant information for the reproduction of the results, including, for instance, the specific hyperparameter values considered in each of the experiments. In addition, we provide access to our code (https://anonymous.4open.science/r/SEMoLA/), which includes the configuration files used in the execution of all the experiments, as well as instructions on how to access the different datasets, and we also plan to publicly release it upon acceptance of the paper.

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

## A  EXPANDED EXPERIMENTAL DETAILS

### A.1  TECHNICAL INFORMATION

All experiments reported in this paper were executed on a single NVIDIA A100 GPU (except for the QM9 dataset, for which we used an H100) using Pytorch (Ansel et al., 2024) and CUDA version 12.2. We used the Adam optimizer with a learning rate of 0.001, keeping the rest of its default hyperparameter values. As a note, we should mention that even though we report the train and test times for each of the experiments, we notice that those results are easily influenced by the utilization of the GPU at the time of their execution, so some variability can be expected across experiments.

### A.2  HYPERPARAMETER CONFIGURATIONS

In order to facilitate a fair comparison of our approach with the considered baselines, we run them using the same exact hyperparameter configurations provided in their papers or in available implementations. Moreover, we make sure to run all the models with the number of epochs and batch size considered in those configurations, instead of defining those values based on the ones that lead to the best performance from our models. Therefore, it is possible that the results of SEMoLA could be slightly improved with a more detailed tuning and selection of such values. Furthermore, we do not carry out an extensive fine-grained hyperparameter search over those hyperparameters unique to our approach, due to the required computational and temporal resources that would entail. Thus, it is possible that this would be another avenue to further improve the performance of our proposal. Lastly, we should note that the hyperparameter values for the base models and base models with fixed augmentations exactly coincide with those of SEMoLA in all of the experiments.

### A.3  ADDITIONAL METRICS

In order to provide a more detailed analysis of the equivariance error incurred by the models, we also report their results in all of the considered experiments for another metric from Elhag et al. (2024):

$$E'(f_\theta, G) = \frac{1}{N} \sum_{i=1}^{N} \frac{1}{K} \sum_{j=1}^{K} \| f_\theta(\rho_\mathcal{X}(g_j)(x_i)) - \rho_\mathcal{Y}(g_j)(f_\theta(x_i)) \|, \tag{4}$$

where, again, $\|\|$ is defined to be the $L_1$ norm, and we consider $K = 10$. We should note that we find this metric to behave very similarly to the one defined in Equation 3, but we include it nonetheless for completeness.

In addition, we also report the total train and test times that correspond to the different prediction models to complement their associated performance metrics. Importantly, those time measurements do not include the additional time required by decoupled symmetry discovery approaches, e.g., LieGAN, for which the prediction models rely on augmentations derived from previously discovered symmetries. That additional runtime for those two-step equivariant prediction approaches can pose a significant limitation in practice, and it is one of the motivations behind our proposal of a single end-to-end method.

### A.4  ROTATEDMNIST

#### A.4.1  EXPERIMENTAL DETAILS

The base model used in SEMoLA, as well as the other baselines that consider it, is defined to be a very simple CNN, with four convolutional filters and two interleaved max-pooling layers. Following Cohen and Welling (2016), the GCNN model is defined as invariant to the $p4$ group, which consists of all possible compositions of translations and rotations by 90 degrees about any center of rotation in a square grid. Its underlying implementation is defined analogously to our base CNN model in terms of number and type of layers, in order to facilitate the comparison. Also, we should note that while this model encodes the invariance into the architecture in a 'hard' manner, its performance may be limited by the fact that the discrete group to which it is invariant is not an exact match to the underlying continuous group of the dataset, so while the equivariant errors should be low, its actual classification performance might be somewhat restricted.

In this experiment, we applied the following hyperparameter values to our model: $\alpha = 1.0$, $\beta = 7.0$, $\lambda = 0.1$, $\eta = 0.0$, $\nu = 0.01$, $\gamma = 3.0$ for the uniform distribution in the LieAugmenter, and number of augmentations $K = 10$. We should note that $\eta$ has no effect on this particular scenario, because the cardinality of the basis is $C = 1$. In addition, we maintain the default train-test splits of MNIST (Deng, 2012) but also create a validation set with $10\%$ of the training set samples, thus having $54,000$ training samples, $6,000$ validation samples, and $10,000$ test samples. We train all models for 15 epochs with a batch size of 64 for three runs with different random seeds and report the mean and standard deviation of the performances.

### A.4.2 DEFAULT INFERENCE STRATEGY

We complement the performance results from the paper (Table 2) with the corresponding total train and test times of the prediction models, together with the second equivariant error (Equation 4) in Table 5. From them, we can see that our model takes the longest time to train, which can mainly be explained by its additional complexity in terms of more trainable parameters or loss terms than the alternatives, as well as for our currently not excessively optimized implementation. In addition, the second equivariant error behaves similarly to the one shown in the paper.

Table 5: Results for the train and test times (MM:SS) and equivariant error from Equation 4 of the different considered models for in-distribution and out-of-distribution versions of the RotatedMNIST dataset. We report the mean of all the metrics and the standard deviation of the equivariant error across three runs using different random seeds.

| Model | In distribution | | | Out of distribution | | |
|---|---|---|---|---|---|---|
| | Train time | Test time | Equiv. Error' | Train time | Test time | Equiv. Error' |
| Augerino+ | $04:30$ | $00:03$ | $6.35_{\pm 0.98}$ | $04:47$ | $00:03$ | $5.26_{\pm 0.72}$ |
| CNN | $01:52$ | $00:00$ | $6.69_{\pm 0.24}$ | $01:50$ | $00:00$ | $8.74_{\pm 0.61}$ |
| CNN + GT aug | $06:55$ | $00:02$ | $4.86_{\pm 0.36}$ | $06:57$ | $00:02$ | $4.75_{\pm 0.38}$ |
| CNN + LieGAN aug | $06:15$ | $00:01$ | $6.12_{\pm 0.25}$ | $05:03$ | $00:01$ | $7.17_{\pm 0.50}$ |
| GCNN | $03:26$ | $00:00$ | $6.51_{\pm 1.41}$ | $03:56$ | $00:00$ | $7.16_{\pm 1.52}$ |
| SEMoLA | $07:47$ | $00:01$ | $3.82_{\pm 0.38}$ | $07:04$ | $00:01$ | $4.12_{\pm 0.75}$ |

### A.4.3 ALTERNATIVE INFERENCE STRATEGY

Furthermore, we report the performance of the alternative inference strategy of producing predictions only for the original samples without any augmentations in Tables 6 and 7, from which we can see that, as expected, the performance slightly decreases with respect to the previous inference strategy, but the total test time is reduced.

### A.4.4 SAMPLE EFFICIENCY

Next, we explore the sample efficiency of SEMoLA when compared to the other considered prediction approaches by measuring their test performance when trained with decreasing fractions of samples in the training set. As we can see in Figure 5, SEMoLA matches the performance of the CNN with ground truth augmentations up to training with 50% of the data, after which its performance decreases but remains higher than that of the other alternatives.

### A.4.5 EMPIRICAL CONVERGENCE

While the difficulty of analyzing the joint minimization problem corresponding to SEMoLA puts an exact theoretical analysis of its convergence beyond the scope of this paper, we carry out an empirical exploration in order to provide greater insight and evidence of the applicability of our proposed approach in practice.

In that sense, we report the train and validation loss curves as well as the corresponding separate evolution of the main loss terms of the train loss (i.e., objective/empirical loss and equivariance loss) during training in Figure 6. As we can observe, all losses quickly decrease in the first five epochs and

Table 6: Results for the performance and equivariant error of the different considered models for in-distribution and out-of-distribution versions of the RotatedMNIST, where inference for SEMoLA and CNN + aug models is computed using the second strategy described in the paper, i.e., not computing any augmentations and simply producing the predictions for the original inputs. For each metric, we report the mean and standard deviation across three runs with different random seeds. The accuracies are expressed as percentages.

| Model | In distribution | | Out of distribution | |
|---|---|---|---|---|
| | Accuracy | Equiv. Error | Accuracy | Equiv. Error |
| Augerino+ | $96.85_{\pm 0.30}$ | $5.09_{\pm 0.70}$ | $57.68_{\pm 1.45}$ | $3.97_{\pm 0.51}$ |
| CNN | $96.17_{\pm 0.34}$ | $5.52_{\pm 0.23}$ | $55.24_{\pm 1.14}$ | $7.19_{\pm 0.54}$ |
| CNN + GT aug | $98.45_{\pm 0.11}$ | $3.91_{\pm 0.23}$ | $98.35_{\pm 0.11}$ | $4.03_{\pm 0.17}$ |
| CNN + LieGAN aug | $97.84_{\pm 0.13}$ | $4.92_{\pm 0.74}$ | $84.97_{\pm 0.43}$ | $6.80_{\pm 0.85}$ |
| GCNN | $96.78_{\pm 0.36}$ | $5.31_{\pm 1.16}$ | $55.60_{\pm 1.45}$ | $5.78_{\pm 1.23}$ |
| SEMoLA | $98.55_{\pm 0.08}$ | $3.72_{\pm 0.15}$ | $93.40_{\pm 2.46}$ | $3.40_{\pm 0.47}$ |

Table 7: Results for the train and test times (MM:SS) and equivariant error from Equation 4 of the different considered models for in-distribution and out-of-distribution versions of the RotatedMNIST dataset, where inference for SEMoLA and CNN + augmentations models is computed using the second strategy described in the paper, i.e., not computing any augmentations and simply producing the predictions for the original inputs. We report the mean of all the metrics and the standard deviation of the equivariant error across three runs using different random seeds.

| Model | In distribution | | | Out of distribution | | |
|---|---|---|---|---|---|---|
| | Train time | Test time | Equiv. Error' | Train time | Test time | Equiv. Error' |
| Augerino+ | $04:30$ | $00:03$ | $6.35_{\pm 0.98}$ | $04:47$ | $00:03$ | $5.26_{\pm 0.72}$ |
| CNN | $01:52$ | $00:00$ | $6.69_{\pm 0.24}$ | $01:50$ | $00:00$ | $8.74_{\pm 0.61}$ |
| CNN + GT aug | $03:03$ | $00:00$ | $4.72_{\pm 0.27}$ | $06:57$ | $00:00$ | $4.96_{\pm 0.13}$ |
| CNN + LieGAN aug | $09:45$ | $00:00$ | $5.95_{\pm 0.87}$ | $06:18$ | $00:00$ | $8.35_{\pm 0.97}$ |
| GCNN | $03:26$ | $00:00$ | $6.51_{\pm 1.41}$ | $03:56$ | $00:00$ | $7.16_{\pm 1.52}$ |
| SEMoLA | $10:44$ | $00:00$ | $4.54_{\pm 0.17}$ | $10:48$ | $00:00$ | $4.29_{\pm 0.56}$ |

then start converging around epoch ten. These observations are also similarly reflected in other runs for this same dataset as well as for other more complex datasets (see, e.g., Appendix A.6).

### A.4.6 EXAMPLE SAMPLED AUGMENTATIONS

Finally, we plot some examples of the augmentations generated by SEMoLA for different digits in Figure 7, which serves to verify that it is indeed learning to generate augmentations that correspond to rotations of the input image and that the rotations are varied across the augmented samples.

### A.5 N-BODY DYNAMICS

### A.5.1 EXPERIMENTAL DETAILS

In this experiment, we used a simple MLP, consisting of four layers and ReLU activations, as the base model for SEMoLA and the other baselines that require it. We consider EMLP Finzi et al. (2021) as the 'hard' equivariant model for this problem, which we define based either on the ground truth Lie algebra basis, or on the bases learned by LieGAN.

We applied the following hyperparameter values to our model: $\alpha = 1.0$, $\beta = 10.0$, $\lambda = 1.0$, $\eta = 0.0$, $\nu = 0.001$, $\gamma = 2.0$ for the uniform distribution in the LieAugmenter, and number of augmentations $K = 10$. Again, $\eta$ has no effect on this particular scenario, because the cardinality of the basis is

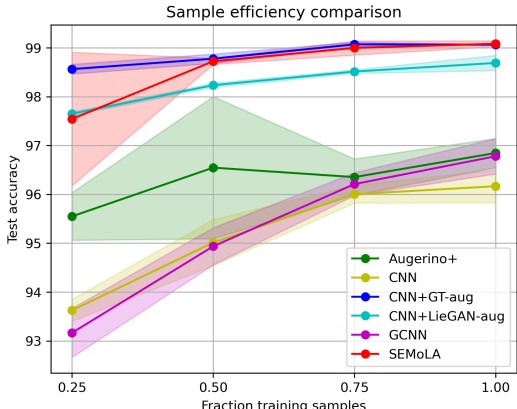

Figure 5: Comparison of the sample efficiency of SEMoLA, the base CNN model, Augerino+, GCNN, and the base CNN model trained with ground truth and LieGAN augmentations, which are trained with different fractions of the total training set samples of the RotatedMNIST dataset.

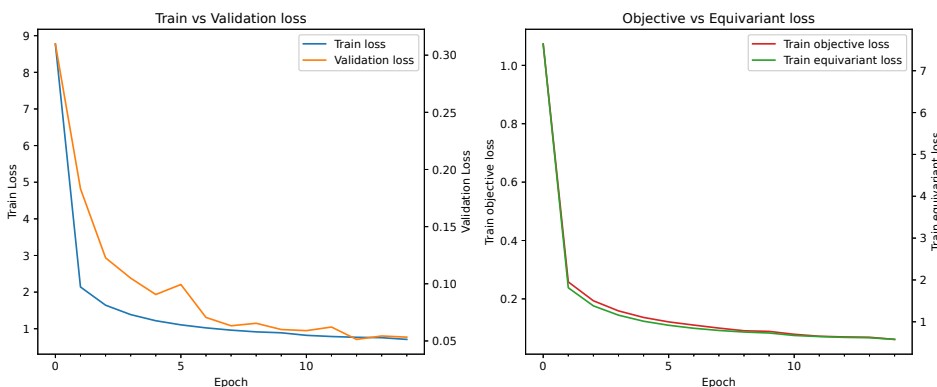

Figure 6: Loss curves of SEMoLA for RotatedMNIST dataset. (Left) Train and validation losses. (Right) Objective and equivariance losses.

$C = 1$. In terms of the dataset, we maintain the default train-test splits defined in Yang et al. (2023). We train all models for 100 epochs with a batch size of 64 for three runs with different random seeds and report the mean and standard deviation of the performances.

As mentioned in the paper, we test the impact of introducing different restrictions into the structure of the learned Lie algebra basis to explore the capabilities of the symmetry discovery approaches under varying degrees of auxiliary knowledge about the task. In that sense, we define different 'masks' that are multiplied by the learned Lie algebra before the default steps of the LieAugmenter in order to restrict the learnable entries being considered in the augmentations. As done in Yang et al. (2023), we consider a mask that implies searching for symmetries acting on the position and momentum of each mass separately (Mask_2), and a mask that allows interactions between only the positions and only the momenta across the different bodies (Mask_4). In addition, we again report the results from the paper in which we do not impose any mask (Mask_0). We add the same numerical suffixes to the considered models to indicate which mask each uses.

### A.5.2   1-TIMESTEP PREDICTION

Firstly, we complement the experimental results presented in the paper for the case of predicting one output timestep from one input timestep.

In that sense, the symmetry discovery results presented in Figure 8 and 8 show that LieGAN (Yang et al., 2023) is able to identify the correct symmetry in the Mask_2 setting of the in-distribution

Figure 7: Example of augmentations generated by SEMoLA for the RotatedMNIST dataset. The first column shows the original samples, and the subsequent ones correspond to the generated augmentations.

scenario, as well as a noisy version of the alternative representation of the symmetry in the Mask_4 setting (see Appendix A.5.5), while Augerino+ (Yang et al., 2023) fails in both cases while producing much noisier results. In the out-of-distribution scenario, Augerino+ (Yang et al., 2023) still fails to learn the correct symmetry, while now LieGAN (Yang et al., 2023) fails to learn the symmetry in the Mask_2 setting. This result exemplifies the sensitivity of the GAN-based symmetry discovery methods to the uniformity of the considered dataset, since they rely on discovering distribution-preserving transformations. Meanwhile, we can see that SEMoLA successfully identifies the correct symmetry (or its alternative representation) under all masks while producing less noisy and sparser representations, which are generally preferable.

Table 8: Cosine and MAE similarities of the ground truth Lie algebra basis and the learned ones by each of the considered symmetry learning models in the in-distribution and out-of-distribution versions of the 2-body dataset with one input and output timestep. For each metric, we report the mean and standard deviation across three runs using different random seeds.

| Model | In distribution | | Out of distribution | |
|---|---|---|---|---|
| | Cosine | MAE | Cosine | MAE |
| Augerino+_0 | $0.2302_{\pm 0.0122}$ | $1.2645_{\pm 0.0900}$ | $0.1974_{\pm 0.1417}$ | $1.0338_{\pm 0.0353}$ |
| Augerino+_2 | $0.3049_{\pm 0.1762}$ | $0.3660_{\pm 0.0790}$ | $0.2927_{\pm 0.1684}$ | $0.3660_{\pm 0.0782}$ |
| Augerino+_4 | $0.6295_{\pm 0.0071}$ | $0.4130_{\pm 0.0914}$ | $0.6421_{\pm 0.0043}$ | $0.7650_{\pm 0.0114}$ |
| LieGAN_0 | $0.2089_{\pm 0.0289}$ | $0.3013_{\pm 0.0185}$ | $0.0717_{\pm 0.0280}$ | $0.2208_{\pm 0.0223}$ |
| LieGAN_2 | $0.9993_{\pm 0.0002}$ | $0.1339_{\pm 0.0222}$ | $0.6154_{\pm 0.1573}$ | $0.1250_{\pm 0.0003}$ |
| LieGAN_4 | $0.6859_{\pm 0.0259}$ | $0.2067_{\pm 0.0517}$ | $0.7010_{\pm 0.0013}$ | $0.2753_{\pm 0.1045}$ |
| SEMoLA_0 | $0.7096_{\pm 0.0097}$ | $0.1873_{\pm 0.0825}$ | $0.7046_{\pm 0.0024}$ | $0.1293_{\pm 0.0008}$ |
| SEMoLA_2 | $1.0000_{\pm 0.0000}$ | $0.0058_{\pm 0.0008}$ | $0.6672_{\pm 0.4706}$ | $0.0506_{\pm 0.0619}$ |
| SEMoLA_4 | $0.7075_{\pm 0.0001}$ | $0.1836_{\pm 0.0827}$ | $0.7071_{\pm 0.0011}$ | $0.1263_{\pm 0.0005}$ |

Moreover, we extend the results from Table 3 with the performance results for the different masking scenarios and their corresponding total train and test times together with the second equivariant error (Equation 4) in Tables 9 and 10, respectively. From these results, we can see that Augerino+ (Yang et al., 2023) displays the worst performance, due to it not being able to learn the correct symmetry,

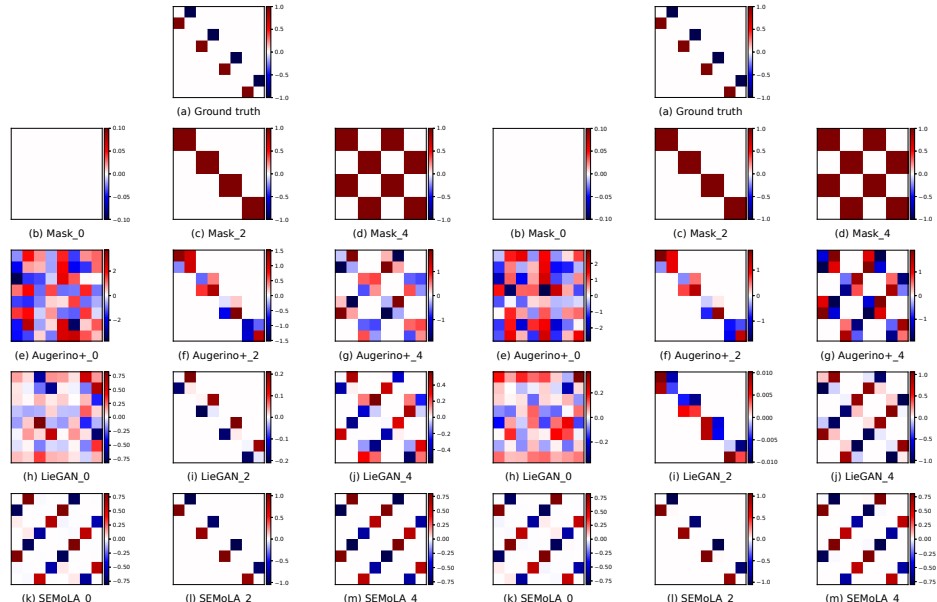

Figure 8: Comparison of the ground truth Lie algebra basis with those learned by the baselines LieGAN and Augerino+, and by our proposed SEMoLA approach for the in-distribution versions of the 2-body dataset (left) and the restricted out-of-distribution version (right) with one input and output timestep.

as previously discussed. The base MLP model has the second-worst performance, since it lacks any knowledge about the underlying equivariance of the data. Next, in terms of the MLP with the different fixed augmentations, we can see that using the ground truth symmetry provides the best performance among them, while using the symmetry discovered by LieGAN only performs well when it uses the Mask_4 setting. The EMLP (Finzi et al., 2021) model provides the best overall performance regardless of the Lie algebra basis considered for its definition, thanks to its 'hard' equivariance design, but notably it requires by far the longest train and test times. Lastly, SEMoLA provides the overall second-best performance regardless of the 'mask', being only worse than EMLP (Finzi et al., 2021) and performing very similarly to the MLP with ground truth augmentations. As a result, it provides the best performance among all prediction models that do not consider any a priori knowledge about the symmetry.

### A.5.3    3-TIMESTEP PREDICTION

Next, we further extend the experimental results presented in the paper with the more complex scenario of predicting three output timesteps from three input timesteps.

We can find the symmetry discovery results in Figure 9 and 11, which are, for the most part, consistent with those previously shown in the 1-timestep case, but slightly noisier. Overall, while Augerino+ (Yang et al., 2023) and LieGAN (Yang et al., 2023) still struggle to learn the correct symmetry in most settings, SEMoLA is able to successfully learn it under all masks.

In terms of the performance results for this 3-timestep prediction setting, we can find them in Tables 12 and 13, where we can again observe very similar trends as in the 1-timestep prediction scenario. In particular, Augerino+ (Yang et al., 2023) and the MLP with LieGAN augmentations perform the worst, while the EMLP and the MLP with ground truth augmentations perform best, closely followed by SEMoLA, which provides the best performance among prediction models that do not consider any a priori knowledge about the symmetry.

Table 9: Results for the performance and equivariant error of the different considered models for in-distribution and out-of-distribution versions of the 2-body dataset with one input and output timestep. For each metric, we report the mean and standard deviation across three runs using different random seeds. The accuracies are expressed as percentages.

| Model | In distribution | | Out of distribution | |
|---|---|---|---|---|
| | MSE | Equiv. Error | MSE | Equiv. Error |
| Augerino+_0 | $5.28e{+}00_{\pm 4.23e\text{-}01}$ | $1.90e\text{-}01_{\pm 1.92e\text{-}03}$ | $3.26e{+}00_{\pm 7.30e\text{-}01}$ | $1.94e\text{-}01_{\pm 4.15e\text{-}03}$ |
| Augerino+_2 | $9.67e{+}03_{\pm 1.37e{+}04}$ | $2.37e\text{-}01_{\pm 7.53e\text{-}02}$ | $1.48e{+}05_{\pm 2.09e{+}05}$ | $1.95e\text{-}01_{\pm 9.23e\text{-}03}$ |
| Augerino+_4 | $1.79e{+}00_{\pm 2.77e\text{-}02}$ | $1.87e\text{-}01_{\pm 2.92e\text{-}04}$ | $1.55e{+}00_{\pm 8.59e\text{-}03}$ | $1.86e\text{-}01_{\pm 5.90e\text{-}04}$ |
| MLP | $5.16e\text{-}03_{\pm 2.79e\text{-}03}$ | $2.48e\text{-}02_{\pm 7.37e\text{-}03}$ | $1.86e\text{-}02_{\pm 3.74e\text{-}03}$ | $3.33e\text{-}02_{\pm 2.56e\text{-}03}$ |
| MLP + GT aug | $2.13e\text{-}05_{\pm 1.56e\text{-}05}$ | $3.03e\text{-}03_{\pm 1.03e\text{-}03}$ | $3.03e\text{-}05_{\pm 8.81e\text{-}06}$ | $3.70e\text{-}03_{\pm 1.14e\text{-}03}$ |
| MLP + LieGAN_0 aug | $3.23e\text{-}03_{\pm 6.25e\text{-}04}$ | $1.91e\text{-}02_{\pm 1.47e\text{-}03}$ | $1.15e\text{-}02_{\pm 3.00e\text{-}03}$ | $2.84e\text{-}02_{\pm 3.04e\text{-}03}$ |
| MLP + LieGAN_2 aug | $1.10e\text{-}03_{\pm 6.98e\text{-}04}$ | $1.48e\text{-}02_{\pm 4.80e\text{-}03}$ | $8.94e\text{-}03_{\pm 9.31e\text{-}04}$ | $2.73e\text{-}02_{\pm 1.10e\text{-}03}$ |
| MLP + LieGAN_4 aug | $2.85e\text{-}05_{\pm 1.19e\text{-}05}$ | $3.02e\text{-}03_{\pm 5.08e\text{-}04}$ | $5.12e\text{-}05_{\pm 8.47e\text{-}06}$ | $2.78e\text{-}03_{\pm 3.62e\text{-}04}$ |
| EMLP + GT | $1.74e\text{-}06_{\pm 1.50e\text{-}06}$ | $5.32e\text{-}05_{\pm 1.16e\text{-}05}$ | $4.64e\text{-}06_{\pm 4.86e\text{-}06}$ | $5.27e\text{-}05_{\pm 1.10e\text{-}05}$ |
| EMLP + LieGAN_0 | $6.36e\text{-}06_{\pm 4.56e\text{-}06}$ | $5.31e\text{-}05_{\pm 1.13e\text{-}05}$ | $3.81e\text{-}06_{\pm 2.62e\text{-}06}$ | $5.31e\text{-}05_{\pm 1.09e\text{-}05}$ |
| EMLP + LieGAN_2 | $4.82e\text{-}06_{\pm 3.19e\text{-}06}$ | $5.33e\text{-}05_{\pm 1.11e\text{-}05}$ | $1.93e\text{-}06_{\pm 1.84e\text{-}06}$ | $5.30e\text{-}05_{\pm 1.14e\text{-}05}$ |
| EMLP + LieGAN_4 | $2.15e\text{-}06_{\pm 1.98e\text{-}06}$ | $5.30e\text{-}05_{\pm 1.16e\text{-}05}$ | $3.29e\text{-}06_{\pm 2.33e\text{-}06}$ | $5.28e\text{-}05_{\pm 1.14e\text{-}05}$ |
| SEMoLA_0 | $1.39e\text{-}05_{\pm 6.08e\text{-}07}$ | $2.78e\text{-}03_{\pm 4.88e\text{-}04}$ | $2.38e\text{-}05_{\pm 9.04e\text{-}06}$ | $2.59e\text{-}03_{\pm 9.89e\text{-}05}$ |
| SEMoLA_2 | $1.30e\text{-}05_{\pm 5.58e\text{-}06}$ | $3.28e\text{-}03_{\pm 9.36e\text{-}04}$ | $3.73e\text{-}05_{\pm 2.97e\text{-}05}$ | $5.96e\text{-}03_{\pm 2.82e\text{-}03}$ |
| SEMoLA_4 | $1.45e\text{-}05_{\pm 6.15e\text{-}06}$ | $3.15e\text{-}03_{\pm 9.78e\text{-}04}$ | $1.61e\text{-}05_{\pm 1.22e\text{-}06}$ | $3.13e\text{-}03_{\pm 2.79e\text{-}04}$ |

Table 10: Results for the train and test times (MM:SS) and equivariant error from Equation 4 of the different considered models for in-distribution and out-of-distribution versions of the 2-body dataset with one input and output timestep. We report the mean of all the metrics and the standard deviation of the equivariant error across three runs using different random seeds.

| Model | In distribution | | | Out of distribution | | |
|---|---|---|---|---|---|---|
| | Train time | Test time | Equiv. Error' | Train time | Test time | Equiv. Error' |
| Augerino+_0 | $05:25$ | $00:02$ | $4.20e\text{-}01_{\pm 1.85e\text{-}03}$ | $05:25$ | $00:02$ | $4.21e\text{-}01_{\pm 6.31e\text{-}03}$ |
| Augerino+_2 | $05:40$ | $00:02$ | $4.30e\text{-}01_{\pm 5.04e\text{-}02}$ | $05:32$ | $00:02$ | $4.00e\text{-}01_{\pm 6.40e\text{-}02}$ |
| Augerino+_4 | $05:33$ | $00:02$ | $4.16e\text{-}01_{\pm 1.64e\text{-}03}$ | $05:30$ | $00:02$ | $4.16e\text{-}01_{\pm 3.58e\text{-}05}$ |
| MLP | $02:00$ | $00:00$ | $4.84e\text{-}02_{\pm 1.58e\text{-}02}$ | $02:01$ | $00:00$ | $8.91e\text{-}02_{\pm 6.24e\text{-}03}$ |
| MLP + GT aug | $02:48$ | $00:00$ | $6.35e\text{-}03_{\pm 1.44e\text{-}03}$ | $03:10$ | $00:00$ | $6.53e\text{-}03_{\pm 1.25e\text{-}03}$ |
| MLP + LieGAN_0 aug | $05:29$ | $00:00$ | $3.93e\text{-}02_{\pm 1.73e\text{-}03}$ | $05:25$ | $00:00$ | $7.25e\text{-}02_{\pm 1.01e\text{-}02}$ |
| MLP + LieGAN_2 aug | $05:19$ | $00:00$ | $2.83e\text{-}02_{\pm 8.22e\text{-}03}$ | $05:15$ | $00:00$ | $6.68e\text{-}02_{\pm 2.19e\text{-}03}$ |
| MLP + LieGAN_4 aug | $05:22$ | $00:00$ | $6.03e\text{-}03_{\pm 8.00e\text{-}04}$ | $05:26$ | $00:00$ | $7.08e\text{-}03_{\pm 1.67e\text{-}03}$ |
| EMLP + GT | $21:34$ | $00:02$ | $8.08e\text{-}05_{\pm 1.76e\text{-}05}$ | $21:13$ | $00:02$ | $8.02e\text{-}05_{\pm 1.72e\text{-}05}$ |
| EMLP + LieGAN_0 | $19:22$ | $00:02$ | $8.06e\text{-}05_{\pm 1.73e\text{-}05}$ | $15:39$ | $00:02$ | $8.08e\text{-}05_{\pm 1.71e\text{-}05}$ |
| EMLP + LieGAN_2 | $21:56$ | $00:02$ | $8.07e\text{-}05_{\pm 1.72e\text{-}05}$ | $21:18$ | $00:02$ | $8.05e\text{-}05_{\pm 1.75e\text{-}05}$ |
| EMLP + LieGAN_4 | $21:36$ | $00:02$ | $8.04e\text{-}05_{\pm 1.78e\text{-}05}$ | $16:09$ | $00:02$ | $8.03e\text{-}05_{\pm 1.76e\text{-}05}$ |
| SEMoLA_0 | $08:18$ | $00:00$ | $5.75e\text{-}03_{\pm 2.26e\text{-}04}$ | $08:20$ | $00:00$ | $4.53e\text{-}03_{\pm 2.52e\text{-}04}$ |
| SEMoLA_2 | $08:28$ | $00:00$ | $5.34e\text{-}03_{\pm 1.35e\text{-}03}$ | $08:26$ | $00:00$ | $9.61e\text{-}03_{\pm 4.93e\text{-}03}$ |
| SEMoLA_4 | $08:27$ | $00:00$ | $6.17e\text{-}03_{\pm 1.61e\text{-}03}$ | $08:26$ | $00:00$ | $5.79e\text{-}03_{\pm 7.28e\text{-}04}$ |

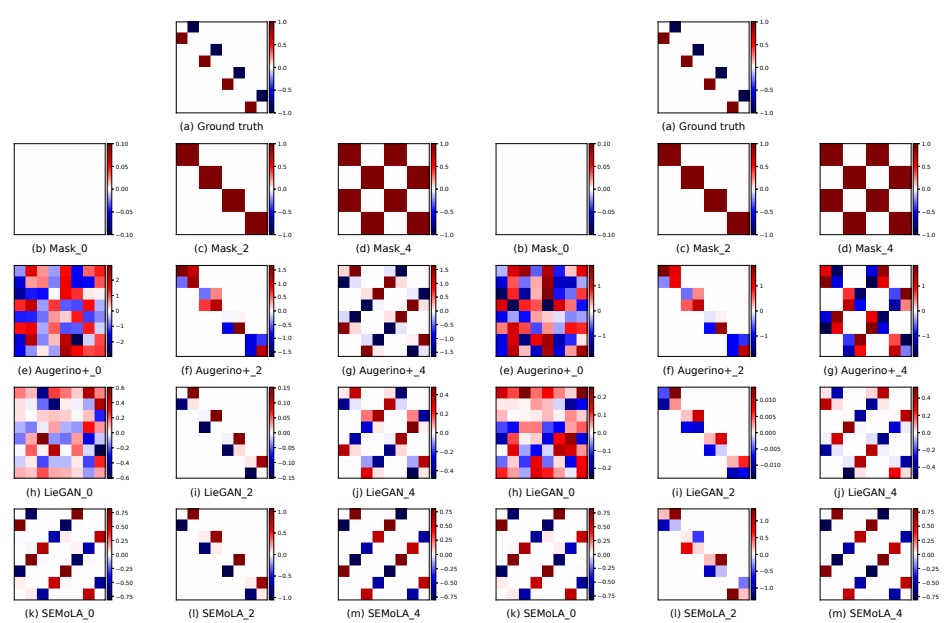

Figure 9: Comparison of the ground truth Lie algebra basis with those learned by the baselines LieGAN and Augerino+, and by our proposed SEMoLA approach for the in-distribution version of the 2-body dataset (left) and the restricted out-of-distribution version (right) with three input and output timesteps.

Table 11: Cosine and MAE similarities of the ground truth Lie algebra basis and the learned ones by each of the considered symmetry learning models in the in-distribution and out-of-distribution versions of the 2-body dataset with three input and output timesteps. For each metric, we report the mean and standard deviation across three runs using different random seeds.

| Model | In distribution | | Out of distribution | |
|---|---|---|---|---|
| | Cosine | MAE | Cosine | MAE |
| Augerino+_0 | $0.1927_{\pm 0.0958}$ | $1.0683_{\pm 0.1016}$ | $0.1127_{\pm 0.1189}$ | $0.9469_{\pm 0.0487}$ |
| Augerino+_2 | $0.4402_{\pm 0.2548}$ | $0.2905_{\pm 0.0268}$ | $0.4001_{\pm 0.3749}$ | $0.3239_{\pm 0.0428}$ |
| Augerino+_4 | $0.6679_{\pm 0.0291}$ | $0.4709_{\pm 0.1025}$ | $0.6131_{\pm 0.0086}$ | $0.5870_{\pm 0.0546}$ |
| LieGAN_0 | $0.1943_{\pm 0.0241}$ | $0.2734_{\pm 0.0030}$ | $0.1317_{\pm 0.0174}$ | $0.2149_{\pm 0.0041}$ |
| LieGAN_2 | $0.9991_{\pm 0.0001}$ | $0.1189_{\pm 0.0176}$ | $0.8732_{\pm 0.0748}$ | $0.1251_{\pm 0.0011}$ |
| LieGAN_4 | $0.6841_{\pm 0.0206}$ | $0.1729_{\pm 0.0431}$ | $0.6857_{\pm 0.0245}$ | $0.1360_{\pm 0.0058}$ |
| SEMoLA_0 | $0.7104_{\pm 0.0008}$ | $0.1860_{\pm 0.0829}$ | $0.5797_{\pm 0.2021}$ | $0.1456_{\pm 0.0302}$ |
| SEMoLA_2 | $0.6665_{\pm 0.4713}$ | $0.2114_{\pm 0.0571}$ | $0.0017_{\pm 0.0015}$ | $0.1482_{\pm 0.0180}$ |
| SEMoLA_4 | $0.7071_{\pm 0.0006}$ | $0.3007_{\pm 0.0003}$ | $0.7089_{\pm 0.0001}$ | $0.1836_{\pm 0.0828}$ |

Table 12: Results for the performance and equivariant error of the different considered models for in-distribution and out-of-distribution versions of the 2-body dataset with three input and output timesteps. For each metric, we report the mean and standard deviation across three runs using different random seeds. The accuracies are expressed as percentages.

| Model | In distribution | | Out of distribution | |
|---|---|---|---|---|
| | MSE | Equiv. Error | MSE | Equiv. Error |
| Augerino+_0 | $4.03\text{e}{+}00_{\pm 3.49\text{e-}01}$ | $1.91\text{e-}01_{\pm 2.17\text{e-}03}$ | $2.68\text{e}{+}00_{\pm 4.94\text{e-}01}$ | $1.98\text{e-}01_{\pm 9.13\text{e-}03}$ |
| Augerino+_2 | $8.70\text{e}{+}00_{\pm 1.00\text{e}{+}01}$ | $1.96\text{e-}01_{\pm 8.81\text{e-}03}$ | $1.09\text{e}{+}01_{\pm 1.33\text{e}{+}01}$ | $1.87\text{e-}01_{\pm 8.77\text{e-}03}$ |
| Augerino+_4 | $1.72\text{e}{+}00_{\pm 1.03\text{e-}01}$ | $1.88\text{e-}01_{\pm 1.83\text{e-}03}$ | $1.64\text{e}{+}00_{\pm 7.75\text{e-}02}$ | $1.88\text{e-}01_{\pm 2.73\text{e-}03}$ |
| MLP | $9.62\text{e-}03_{\pm 4.32\text{e-}03}$ | $3.58\text{e-}02_{\pm 7.57\text{e-}03}$ | $1.45\text{e-}02_{\pm 1.30\text{e-}03}$ | $3.51\text{e-}02_{\pm 1.39\text{e-}03}$ |
| MLP + GT aug | $3.96\text{e-}05_{\pm 1.75\text{e-}05}$ | $5.08\text{e-}03_{\pm 1.19\text{e-}03}$ | $4.92\text{e-}05_{\pm 2.25\text{e-}05}$ | $3.88\text{e-}03_{\pm 6.66\text{e-}04}$ |
| MLP + LieGAN_0 aug | $8.83\text{e-}03_{\pm 2.72\text{e-}03}$ | $3.59\text{e-}02_{\pm 5.83\text{e-}03}$ | $1.88\text{e-}02_{\pm 4.80\text{e-}03}$ | $4.15\text{e-}02_{\pm 5.98\text{e-}03}$ |
| MLP + LieGAN_2 aug | $2.71\text{e-}03_{\pm 2.39\text{e-}04}$ | $2.26\text{e-}02_{\pm 1.16\text{e-}03}$ | $1.83\text{e-}02_{\pm 4.37\text{e-}03}$ | $4.05\text{e-}02_{\pm 3.98\text{e-}03}$ |
| MLP + LieGAN_4 aug | $8.86\text{e-}05_{\pm 7.58\text{e-}05}$ | $4.24\text{e-}03_{\pm 1.10\text{e-}03}$ | $7.94\text{e-}05_{\pm 2.55\text{e-}05}$ | $4.78\text{e-}03_{\pm 7.64\text{e-}04}$ |
| EMLP + GT | $4.54\text{e-}06_{\pm 2.20\text{e-}06}$ | $2.91\text{e-}05_{\pm 4.71\text{e-}06}$ | $6.30\text{e-}06_{\pm 3.68\text{e-}06}$ | $2.90\text{e-}05_{\pm 4.79\text{e-}06}$ |
| EMLP + LieGAN_0 | $2.99\text{e-}06_{\pm 1.83\text{e-}06}$ | $2.91\text{e-}05_{\pm 4.85\text{e-}06}$ | $5.63\text{e-}06_{\pm 2.62\text{e-}06}$ | $2.91\text{e-}05_{\pm 4.62\text{e-}06}$ |
| EMLP + LieGAN_2 | $2.94\text{e-}06_{\pm 1.21\text{e-}06}$ | $2.91\text{e-}05_{\pm 4.75\text{e-}06}$ | $8.04\text{e-}06_{\pm 5.55\text{e-}06}$ | $2.92\text{e-}05_{\pm 4.66\text{e-}06}$ |
| EMLP + LieGAN_4 | $3.26\text{e-}06_{\pm 1.72\text{e-}06}$ | $2.92\text{e-}05_{\pm 4.77\text{e-}06}$ | $6.40\text{e-}06_{\pm 6.40\text{e-}06}$ | $2.92\text{e-}05_{\pm 4.68\text{e-}06}$ |
| SEMoLA_0 | $5.98\text{e-}05_{\pm 2.17\text{e-}05}$ | $4.07\text{e-}03_{\pm 1.50\text{e-}03}$ | $1.59\text{e-}04_{\pm 1.93\text{e-}04}$ | $4.90\text{e-}03_{\pm 1.88\text{e-}03}$ |
| SEMoLA_2 | $1.75\text{e-}05_{\pm 6.22\text{e-}06}$ | $4.58\text{e-}03_{\pm 4.07\text{e-}04}$ | $3.67\text{e-}04_{\pm 1.10\text{e-}04}$ | $7.97\text{e-}03_{\pm 1.45\text{e-}03}$ |
| SEMoLA_4 | $2.85\text{e-}05_{\pm 3.92\text{e-}06}$ | $3.97\text{e-}03_{\pm 6.82\text{e-}04}$ | $5.32\text{e-}05_{\pm 1.28\text{e-}05}$ | $4.22\text{e-}03_{\pm 1.75\text{e-}04}$ |

Table 13: Results for the train and test times (MM:SS) and equivariant error from Equation 4 of the different considered models for in-distribution and out-of-distribution versions of the 2-body dataset with three input and output timesteps. We report the mean of all the metrics and the standard deviation of the equivariant error across three runs using different random seeds.

| Model | In distribution | | | Out of distribution | | |
|---|---|---|---|---|---|---|
| | Train time | Test time | Equiv. Error' | Train time | Test time | Equiv. Error' |
| Augerino+_0 | 04 : 20 | 00 : 01 | $4.22\text{e-}01_{\pm 3.07\text{e-}03}$ | 04 : 20 | 00 : 01 | $4.22\text{e-}01_{\pm 8.34\text{e-}03}$ |
| Augerino+_2 | 04 : 20 | 00 : 01 | $4.13\text{e-}01_{\pm 1.26\text{e-}02}$ | 04 : 23 | 00 : 01 | $4.13\text{e-}01_{\pm 1.53\text{e-}03}$ |
| Augerino+_4 | 04 : 21 | 00 : 01 | $4.19\text{e-}01_{\pm 4.20\text{e-}03}$ | 04 : 25 | 00 : 01 | $4.20\text{e-}01_{\pm 3.19\text{e-}03}$ |
| MLP | 01 : 36 | 00 : 00 | $7.17\text{e-}02_{\pm 1.53\text{e-}02}$ | 01 : 32 | 00 : 00 | $8.36\text{e-}02_{\pm 3.20\text{e-}03}$ |
| MLP + GT aug | 02 : 34 | 00 : 00 | $1.10\text{e-}02_{\pm 3.29\text{e-}03}$ | 02 : 33 | 00 : 00 | $8.62\text{e-}03_{\pm 8.36\text{e-}04}$ |
| MLP + LieGAN_0 aug | 04 : 22 | 00 : 00 | $7.20\text{e-}02_{\pm 1.13\text{e-}02}$ | 04 : 21 | 00 : 00 | $9.55\text{e-}02_{\pm 1.45\text{e-}02}$ |
| MLP + LieGAN_2 aug | 04 : 16 | 00 : 00 | $4.36\text{e-}02_{\pm 2.40\text{e-}03}$ | 04 : 16 | 00 : 00 | $9.53\text{e-}02_{\pm 1.06\text{e-}02}$ |
| MLP + LieGAN_4 aug | 04 : 17 | 00 : 00 | $8.10\text{e-}03_{\pm 1.21\text{e-}03}$ | 04 : 13 | 00 : 00 | $9.62\text{e-}03_{\pm 3.32\text{e-}03}$ |
| EMLP + GT | 17 : 32 | 00 : 02 | $4.47\text{e-}05_{\pm 7.24\text{e-}06}$ | 16 : 55 | 00 : 02 | $4.46\text{e-}05_{\pm 7.31\text{e-}06}$ |
| EMLP + LieGAN_0 | 16 : 57 | 00 : 02 | $4.48\text{e-}05_{\pm 7.46\text{e-}06}$ | 17 : 10 | 00 : 02 | $4.47\text{e-}05_{\pm 7.16\text{e-}06}$ |
| EMLP + LieGAN_2 | 17 : 05 | 00 : 02 | $4.47\text{e-}05_{\pm 7.33\text{e-}06}$ | 17 : 26 | 00 : 02 | $4.47\text{e-}05_{\pm 7.19\text{e-}06}$ |
| EMLP + LieGAN_4 | 16 : 50 | 00 : 02 | $4.49\text{e-}05_{\pm 7.39\text{e-}06}$ | 15 : 15 | 00 : 02 | $4.47\text{e-}05_{\pm 7.20\text{e-}06}$ |
| SEMoLA_0 | 06 : 37 | 00 : 00 | $7.62\text{e-}03_{\pm 1.95\text{e-}03}$ | 06 : 35 | 00 : 00 | $1.04\text{e-}02_{\pm 4.38\text{e-}03}$ |
| SEMoLA_2 | 06 : 43 | 00 : 00 | $8.50\text{e-}03_{\pm 8.16\text{e-}04}$ | 06 : 40 | 00 : 00 | $2.13\text{e-}02_{\pm 2.66\text{e-}03}$ |
| SEMoLA_4 | 06 : 46 | 00 : 00 | $8.37\text{e-}03_{\pm 9.29\text{e-}04}$ | 06 : 48 | 00 : 00 | $8.11\text{e-}03_{\pm 3.96\text{e-}04}$ |

### A.5.4 EXAMPLE SAMPLED AUGMENTATIONS

We again illustrate some examples of the augmentations generated by SEMoLA for a particular input sample in Figure 10, where we can see that both the input and its corresponding label are properly rotated as corresponds to the learned rotational equivariance.

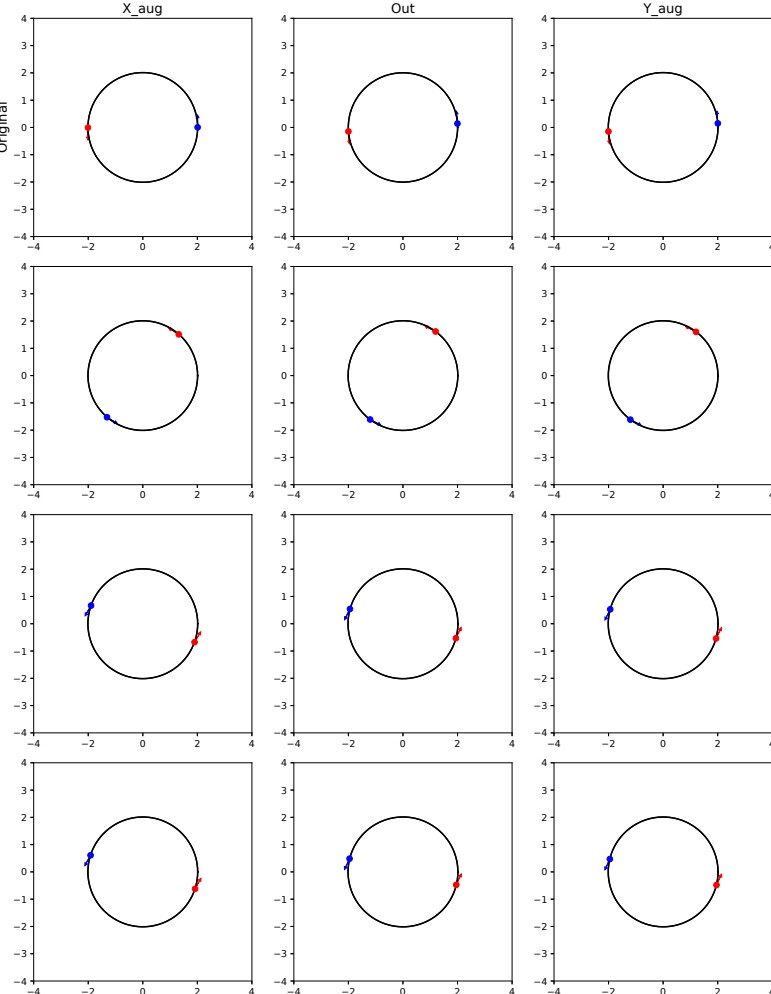

Figure 10: Examples of augmentations generated by SEMoLA for the 2-body dataset. The top row shows (from left to right) the original input sample, the output of the model for that sample, and the target prediction. The rows below show different augmented versions.

### A.5.5 ALTERNATIVE REPRESENTATION OF THE ROTATION SYMMETRY

Throughout our experiments in this dataset, we have observed what we claim to be an alternative representation of the correct symmetry in the context of this 2-body dataset. In that sense, following Yang et al. (2023), we present the derivation that confirms that claim.

Firstly, we can write this alternative representation of the Lie algebra basis in the following way to highlight its structure after removing the slightly noisy entries:

$$R = \begin{bmatrix} 0 & -1 \\ 1 & 0 \end{bmatrix}$$

$$L = \begin{bmatrix} R & & -R & \\ & R & & -R \\ -R & & R & \\ & -R & & R \end{bmatrix}$$

Then, we can express the result of applying the matrix exponential to a weighted version of that basis as follows:

$$\exp(wL) = I + \begin{bmatrix} G(w) & & -G(w) & \\ & G(w) & & -G(w) \\ -G(w) & & G(w) & \\ & -G(w) & & G(w) \end{bmatrix}$$

$$G(w) = \sum_{n=0}^{\infty} \frac{(-1)^n 2^{2n} w^{2n+1}}{(2n+1)!} R + \sum_{n=1}^{\infty} \frac{(-1)^n 2^{2n-1} w^{2n}}{(2n)!} I$$

As a result, we can define its action on the input data as follows:

$$\exp(wL) \begin{bmatrix} \mathbf{q}_1 \\ \mathbf{p}_1 \\ \mathbf{q}_2 \\ \mathbf{p}_2 \end{bmatrix} = \mathrm{diag}(I + 2G(w)) \begin{bmatrix} \mathbf{q}_1 \\ \mathbf{p}_1 \\ \mathbf{q}_2 \\ \mathbf{p}_2 \end{bmatrix},$$

where we have used our knowledge about the dataset, i.e., the origin is at the center of mass, and we have $m_1 = m_2$, $\mathbf{q_1} = -\mathbf{q}_2$, and $\mathbf{p_1} = -\mathbf{p}_2$.

Lastly, we can notice the following relation between the resulting expression and the ground truth rotation symmetry:

$$I + 2G(w) = \sum_{n=0}^{\infty} \frac{(-1)^n 2^{2n+1} w^{2n+1}}{(2n+1)!} R + \sum_{n=0}^{\infty} \frac{(-1)^n 2^{2n} w^{2n}}{(2n)!} I$$

$$= \begin{bmatrix} \cos 2w & -\sin 2w \\ \sin 2w & \cos 2w \end{bmatrix}$$

### A.6  QM9

For our experiments using the QM9 dataset, we set the following hyperparameter values: $\alpha = 1.0$, $\beta = 3.0$, $\lambda = 0.1$, $\eta = 1.0$, $\nu = 0.001$, $\gamma = 3.0$ for the uniform distribution in the LieAugmenter, and number of augmentations $K = 3$. In addition, we maintain the default train-test splits of the dataset (Blum and Reymond, 2009; Rupp et al., 2012). We train SEMoLA for 500 epochs with a batch size of 125. We should note that the reported results from Benton et al. (2020) and Yang et al. (2023) used a slightly different setup, e.g., with the batch size being 75 and the inclusion of a learning rate scheduling strategy. Nevertheless, we believe that, despite those differences, the obtained results are comparable.

In order to complement the results presented in the paper (Table 4), we also report the train and test times and the second equivariant error in Table 14.

In addition, we build on our previous qualitative analysis of the correctness of the learned Lie algebra bases by also computing a quantitative metric that takes into account the mismatched position of the elements of the ground truth and learned bases. In that sense, we compute the average cosine similarity across basis elements between the ground truth Lie algebra basis and different permutations

Table 14: Results for the time (HH:MM:SS) and equivariant error from Equation 4 of SEMoLA for the different considered prediction targets of the QM9 dataset.

| | HOMO | | | LUMO | | |
|---|---|---|---|---|---|---|
| Model | Train time | Test time | Equiv. Error' | Train time | Test time | Equiv. Error' |
| SEMoLA | $15:17:12$ | $00:00:04$ | 0.0045 | $15:19:15$ | $00:00:04$ | 0.0019 |

of the elements of the learned basis, and report the permutation that matches the elements in the ground truth basis in a way that maximizes the similarity for each of the prediction tasks in Table 15. As we can see, the learned Lie algebra bases exactly match the ground truth for this metric after being permuted for both prediction tasks.

Table 15: Maximum average cosine similarity and its corresponding permutation of the elements of the learned Lie algebra basis for each of the considered prediction tasks in the QM9 dataset.

| | Maximum average cosine similarity (Permutation) |
|---|---|
| HOMO | 1.0000 (3, 1, 0, 2, 5, 4) |
| LUMO | 1.0000 (4, 0, 1, 5, 2, 3) |

Lastly, we also carry out an empirical analysis of the convergence of the model for this more complex task through the loss curves shown in Figure 11, where we can again observe the quick convergence of both the equivariance and objective losses despite the complexity of the associated optimization problem.

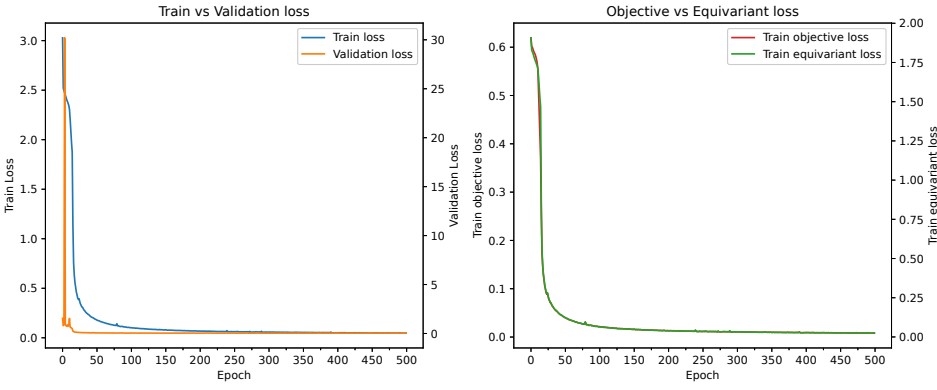

Figure 11: Loss curves for LUMO task of the QM9 dataset. (Left) Train and validation losses. (Right) Objective and equivariance losses.

### A.7 COLORECTAL CANCER DETECTION

As another more complex scenario for analyzing the performance of our proposal, we consider the task of human colorectal cancer (CRC) detection from stained histological images of human tissues. In particular, we consider the datasets NCT-CRC-HE-100K for training and CRC-VAL-HE-7K for testing (Kather et al., 2018), each containing 100,000 and 7,000 samples, respectively, of histological slices of human colorectal cancer and normal tissue, represented as 224x224 RGB images. Rotational invariance has been shown to be particularly important for this classification task (Gerken and Kessel, 2024).

The main goal of this experiment is to explore the applicability of our approach to a more complex and relevant computer vision scenario, where the images are larger and in color. In addition, we also seek to study the effect of applying SEMoLA to more complex architectures besides a simple CNN, e.g., Vision Transformer (ViT) (Dosovitskiy et al., 2020) and EfficientNet (Tan and Le, 2019).

We should note that in this experiment, we do not seek to attain the highest possible performance for the considered models in this dataset, but rather observe the effect of applying SEMoLA in contrast to using ground truth augmentations on each of the base models. As a result, we do not tune the hyperparameters, which we set similarly to previous experiments: $\alpha = 1.0$, $\beta = 7.0$, $\lambda = 0.1$, $\eta = 0.0$, $\nu = 0.01$, $\gamma = 3.0$ for the uniform distribution in the LieAugmenter, and number of augmentations $K = 10$. We train all models for 25 epochs with a batch size of 64.

In terms of symmetry discovery, we follow the same procedure as for the RotatedMNIST dataset and set the cardinality of the Lie algebra basis to $C = 1$ while searching over the six-dimensional space of 2D affine transformations, which are defined to act on the pixel coordinates of the images. With that definition, SEMoLA is able to learn the correct symmetry for all of the considered architectures, as shown in Figure 12.

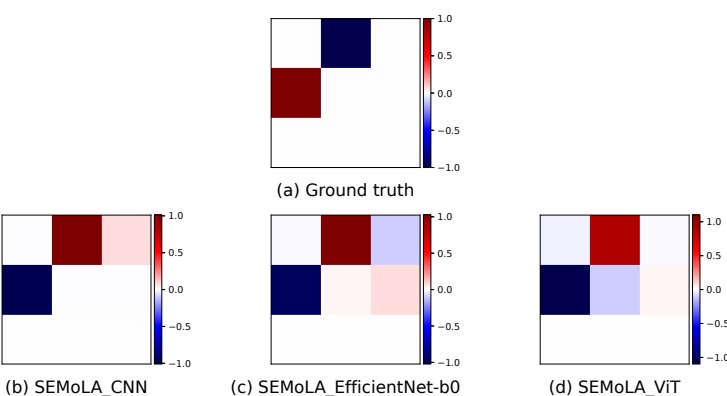

(a) Ground truth

(b) SEMoLA_CNN     (c) SEMoLA_EfficientNet-b0     (d) SEMoLA_ViT

Figure 12: Comparison of the ground truth Lie algebra basis with those learned by the SEMoLA for each of the considered base models in the CRC dataset.

As we can see in the results in Table 16, SEMoLA provides an equal or better performance than using ground truth augmentations for all three models. This observation coincides with and reinforces our previous results, demonstrating once more the benefits of our approach.

Table 16: Results for the performance and equivariant error of the different models considered in the CRC dataset. For each metric, we report the mean and standard deviation across three runs using different random seeds. The accuracies are expressed as percentages.

| Model | Accuracy | Equiv. Error |
|---|---|---|
| CNN | $84.87_{\pm 4.44}$ | $2.95_{\pm 0.15}$ |
| CNN + GT aug | $85.27_{\pm 3.34}$ | $1.75_{\pm 0.29}$ |
| SEMoLA (CNN) | $89.42_{\pm 0.89}$ | $1.86_{\pm 0.21}$ |
| EfficientNet-b0 | $93.04_{\pm 1.18}$ | $2.55_{\pm 0.24}$ |
| EfficientNet-b0 + GT aug | $93.35_{\pm 1.93}$ | $1.49_{\pm 0.14}$ |
| SEMoLA (EfficientNet-b0) | $92.74_{\pm 2.55}$ | $1.21_{\pm 0.10}$ |
| ViT | $75.69_{\pm 3.90}$ | $2.02_{\pm 0.42}$ |
| ViT + GT aug | $81.16_{\pm 2.67}$ | $1.09_{\pm 0.07}$ |
| SEMoLA (ViT) | $85.63_{\pm 2.93}$ | $0.82_{\pm 0.15}$ |

Finally, we also show some examples of the augmentations generated by SEMoLA for each label of the CRC dataset in Figure 13.

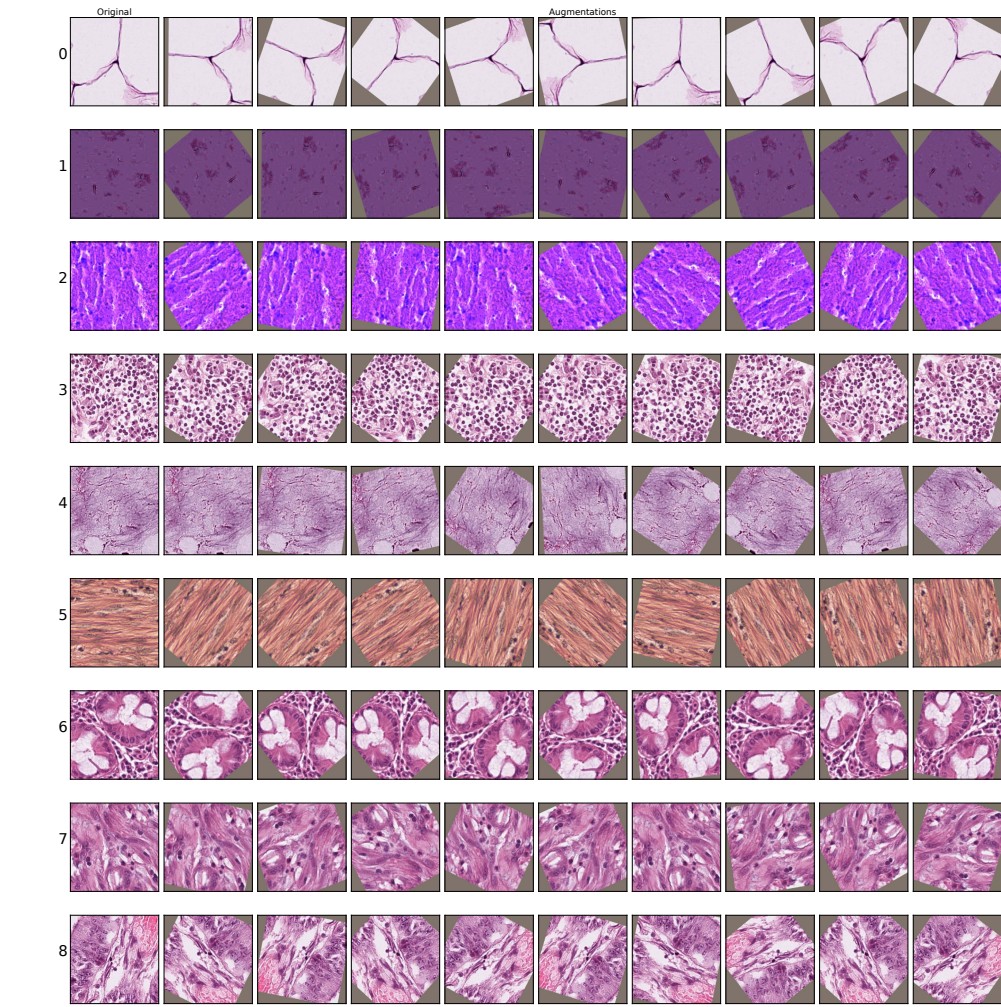

Figure 13: Example of augmentations generated by SEMoLA for the CRC dataset. The first column shows the original samples, and the subsequent ones correspond to the generated augmentations.

### A.8 SYNTHETIC DATASETS

Next, we also explored the ability of SEMoLA to identify the correct Lie group when the symmetry of the data corresponds to discrete subgroups of a Lie group instead of continuous groups, as we explored in the main experiments. To that effect, we construct two synthetic regression datasets based on some scenarios explored in Yang et al. (2023).

For both of these datasets, we set the following hyperparameter values for our model: $\alpha = 1.0$, $\beta = 5.0$, $\lambda = 0.1$, $\eta = 0.0$, $\nu = 0.001$, $\gamma = 3.0$ for the uniform distribution in the LieAugmenter, and number of augmentations $K = 10$. Once more, $\eta$ has no effect on this particular scenario, because the cardinality of the bases is $C = 1$. We generate $54,000$ samples for the training set, $6,000$ for the validation set, and $10,000$ for the test set.

In order to attempt to model the discrete nature of the studied groups, we modify the sampling strategy of the coefficients of the linear combination of the elements of the Lie algebra basis by sampling from a uniform integer grid instead of a uniform distribution. Since prior knowledge about the discrete or continuous nature of the underlying group is not always readily available, we also explore the performance of SEMoLA with the previous uniform sampling strategy to show its robustness to such a possible misspecification. We distinguish the results by the nomenclature 'SEMoLA (discrete)' and 'SEMoLA (continuous)'. Overall, we observe that both sampling strategies provide similarly strong

performance (especially for symmetry discovery), with the discrete formulation providing slightly higher task performance in our experiments.

### A.8.1 DISCRETE ROTATION

As the first synthetic dataset, we consider the synthetic regression problem given by $f(x, y, z) = z/(1 + \arctan \frac{y}{x} \mod \frac{2\pi}{k})$, which is invariant to rotations of multiples of $2\pi/k$ in the $xy$ plane, thus corresponding to a discrete cyclic subgroup of SO(2) with size $k$. Specifically, we consider the case where $k = 6$ and $x, y, z \sim \mathcal{N}(0, I)$.

From Figure 14 and Table 17, we can see that SEMoLA is capable of learning the correct Lie group for the underlying discrete symmetry, while Augerino+(Yang et al., 2023) fails and LieGAN (Yang et al., 2023) provides a much noisier representation.

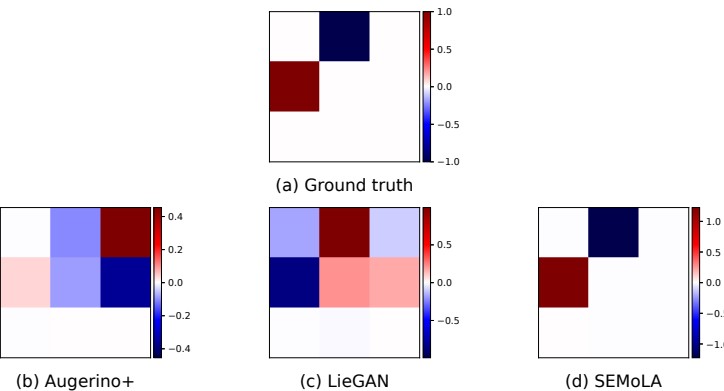

(a) Ground truth

(b) Augerino+        (c) LieGAN        (d) SEMoLA

Figure 14: Comparison of the ground truth Lie algebra basis with those learned by the baselines Augerino+ and LieGAN and our proposed SEMoLA (continuous) in the DiscreteRotation6 dataset.

Table 17: Cosine and MAE similarities of the ground truth Lie algebra basis with those learned by each of the considered symmetry learning models in the DiscreteRotation6 dataset. For each metric, we report the mean and standard deviation across three runs using different random seeds.

| Model | In distribution | |
| --- | --- | --- |
| | Cosine | MAE |
| Augerino+ | $0.4232_{\pm 0.4080}$ | $0.3863_{\pm 0.0581}$ |
| LieGAN | $0.6577_{\pm 0.0095}$ | $0.4419_{\pm 0.0432}$ |
| SEMoLA (continuous) | $1.0000_{\pm 0.0000}$ | $0.1989_{\pm 0.2092}$ |
| SEMoLA (discrete) | $0.8588_{\pm 0.0017}$ | $0.2998_{\pm 0.2098}$ |

In terms of the performance of the different models, we can see in Tables 18 and 19 that SEMoLA (continuous), the MLP with ground truth and LieGAN augmentations, and Augerino+(Yang et al., 2023) all provide extremely similar performance, while the base MLP performs best due to this being a very simple problem for which encoding invariance is not strictly necessary, and SEMoLA (discrete) provides the second best performance.

### A.8.2 PARTIAL PERMUTATION

Next, we seek to show that SEMoLA is able to learn symmetry groups beyond those concerning rotations. In particular, we consider the function $f(x) = x_1 + x_2 + x_3 + x_4^2 - x_5^2$, $x \in \mathbb{R}^5$, which is invariant to partial permutations, with the output remaining the same if we permute the first three dimensions of the input, but changes if we permute the last two. Specifically, we sample $x \sim \mathcal{U}[-100, 100]$.

Table 18: Results for the performance and equivariant error of the different considered models in the DiscreteRotation6 dataset. For each metric, we report the mean and standard deviation across three runs using different random seeds.

| Model | In distribution | |
| --- | --- | --- |
| | MSE | Equiv. Error |
| Augerino+ | 2.09e-02$_{\pm5.56e\text{-}04}$ | 1.08e-02$_{\pm4.22e\text{-}03}$ |
| MLP | 4.06e-03$_{\pm3.02e\text{-}04}$ | 9.57e-02$_{\pm1.54e\text{-}03}$ |
| MLP + GT aug | 2.10e-02$_{\pm1.19e\text{-}03}$ | 2.31e-03$_{\pm2.36e\text{-}04}$ |
| MLP + LieGAN aug | 2.10e-02$_{\pm1.10e\text{-}03}$ | 3.28e-03$_{\pm3.45e\text{-}04}$ |
| SEMoLA (continuous) | 2.12e-02$_{\pm1.24e\text{-}03}$ | 2.46e-03$_{\pm3.29e\text{-}04}$ |
| SEMoLA (discrete) | 3.36e-03$_{\pm4.56e\text{-}04}$ | 3.09e-02$_{\pm1.80e\text{-}03}$ |

Table 19: Results for the train and test times (MM:SS) and equivariant error from Equation 4 of the different considered models for the DiscreteRotation6 dataset. We report the mean of all the metrics and the standard deviation of the equivariant error across three runs using different random seeds.

| Model | In distribution | | |
| --- | --- | --- | --- |
| | Train time | Test time | Equiv. Error' |
| Augerino+ | $00:09:29$ | $00:00:01$ | 1.36e-02$_{\pm5.32e\text{-}03}$ |
| MLP | $00:03:31$ | $00:00:00$ | 1.24e-01$_{\pm2.39e\text{-}03}$ |
| MLP + GT aug | $00:05:35$ | $00:00:00$ | 2.91e-03$_{\pm2.35e\text{-}04}$ |
| MLP + LieGAN aug | $00:09:21$ | $00:00:00$ | 4.21e-03$_{\pm4.20e\text{-}04}$ |
| SEMoLA (continuous) | $00:14:24$ | $00:00:00$ | 3.18e-03$_{\pm4.07e\text{-}04}$ |
| SEMoLA (discrete) | $00:14:51$ | $00:00:00$ | 4.41e-02$_{\pm2.06e\text{-}03}$ |

From Figure 15 and Table 20, we can see that SEMoLA is capable of learning the correct Lie group for the underlying discrete symmetry, while Augerino+(Yang et al., 2023) and LieGAN (Yang et al., 2023) both fail. We should note, however, that in Yang et al. (2023), the authors report that LieGAN (Yang et al., 2023) is able to learn this symmetry, but since the exact hyperparameter values they used are not explicitly stated, we could not reproduce those results exactly.

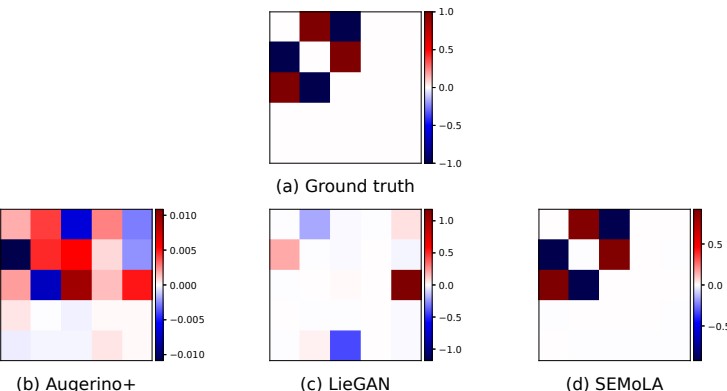

Figure 15: Comparison of the ground truth Lie algebra basis with those learned by the baselines Augerino+ and LieGAN and our proposed SEMoLA (continuous) in the PartialPermutation dataset.

Table 20: Cosine and MAE similarities of the ground truth Lie algebra basis with those learned by each of the considered symmetry learning models in the PartialPermutation dataset. For each metric, we report the mean and standard deviation across three runs using different random seeds.

| | In distribution | |
| Model | Cosine | MAE |
| --- | --- | --- |
| Augerino+ | $0.2890_{\pm 0.3236}$ | $0.2411_{\pm 0.0009}$ |
| LieGAN | $0.1818_{\pm 0.1255}$ | $0.3315_{\pm 0.0024}$ |
| SEMoLA (continuous) | $0.9998_{\pm 0.0002}$ | $0.3140_{\pm 0.2069}$ |
| SEMoLA (discrete) | $0.9987_{\pm 0.0006}$ | $0.3170_{\pm 0.2045}$ |

In terms of the performance of the different models, we can see in Tables 21 and 22 that SEMoLA and the MLP with ground truth augmentations perform best, while the other approaches all produce considerably higher errors.

Table 21: Results for the performance and equivariant error of the different considered models for the PartialPermutation dataset. For each metric, we report the mean and standard deviation across three runs using different random seeds. The accuracies are expressed as percentages.

| | In distribution | |
| Model | MSE | Equiv. Error |
| --- | --- | --- |
| Augerino+ | $3.38e+03_{\pm 8.96e+02}$ | $2.52e+01_{\pm 4.52e+00}$ |
| MLP | $1.97e+03_{\pm 1.20e+03}$ | $1.23e+01_{\pm 3.21e+00}$ |
| MLP + GT aug | $1.69e+03_{\pm 8.44e+02}$ | $1.41e+01_{\pm 2.06e+00}$ |
| MLP + LieGAN aug | $1.24e+04_{\pm 2.90e+03}$ | $1.05e+01_{\pm 2.12e+00}$ |
| SEMoLA (continuous) | $1.77e+03_{\pm 3.57e+02}$ | $1.37e+01_{\pm 4.55e+00}$ |
| SEMoLA (discrete) | $1.49e+03_{\pm 6.55e+01}$ | $1.19e+01_{\pm 2.76e+00}$ |

Table 22: Results for the train and test times (MM:SS) and equivariant error from Equation 4 of the different considered models for the PartialPermutation dataset. We report the mean of all the metrics and the standard deviation of the equivariant error across three runs using different random seeds.

| | In distribution | | |
| Model | Train time | Test time | Equiv. Error' |
| --- | --- | --- | --- |
| Augerino+ | $00:09:55$ | $00:00:01$ | $3.20e+01_{\pm 5.35e+00}$ |
| MLP | $00:03:40$ | $00:00:00$ | $1.58e+01_{\pm 4.25e+00}$ |
| MLP + GT aug | $00:05:33$ | $00:00:00$ | $1.80e+01_{\pm 2.68e+00}$ |
| MLP + LieGAN aug | $00:09:54$ | $00:00:00$ | $1.34e+01_{\pm 2.59e+00}$ |
| SEMoLA (continuous) | $00:15:09$ | $00:00:00$ | $1.75e+01_{\pm 5.80e+00}$ |
| SEMoLA (discrete) | $00:15:04$ | $00:00:00$ | $1.49e+01_{\pm 3.45e+00}$ |

## A.9 ABLATIONS

Lastly, we perform a series of ablations of the different loss weights and other hyperparameters of SEMoLA in order to show their influence on the performance. To that effect, we again consider the RotatedMNIST dataset, which has the fastest execution time among the considered datasets, thus allowing for a more extensive analysis.

From Figure 16 (left), we observe that the performance of our model is quite robust to changes in the values of the weights for the different terms of the loss function except for $\beta$, i.e., the weight for the $\mathcal{L}_{equiv}$ term. In particular, decreasing the value of $\beta$ leads to a degraded performance, as expected, since the LieAugmenter will not receive a strong enough learning signal to properly learn the correct symmetry. In addition, in Figure 16 (right), we can see that the number of sampled augmentations per input sample can also considerably affect the performance of the model, although not so drastically. In particular, increasing the number of augmentations generally leads to an improved performance, as we could expect, but after a certain point, the performance starts decreasing and becomes more variable across runs. One possible reason for that behavior is that at that point the number of augmented samples overcomes the importance of the original samples (due to the number of augmentations per sample being a fourth of the batch size), and the learning behavior becomes impaired due to the model focusing too much on the augmented samples over the original ones.

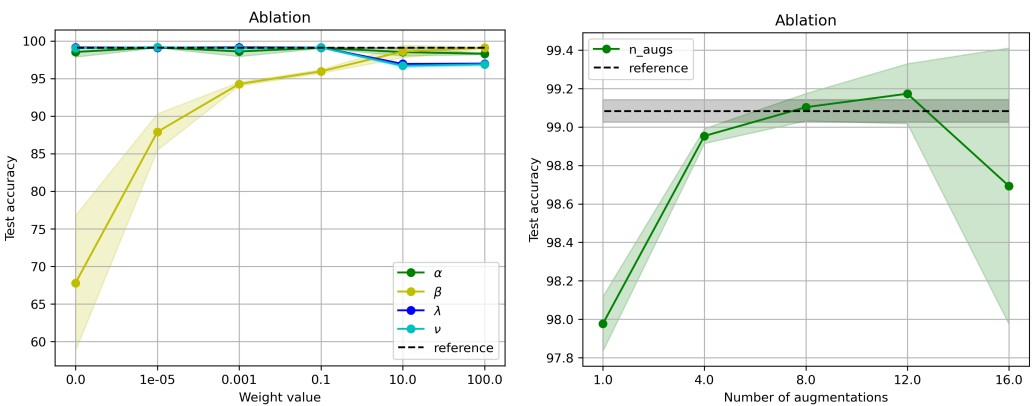

Figure 16: Ablation results for the accuracy metric in the RotatedMNIST dataset for varying values for the weight terms of the loss function (left) and the number of augmentations per sample (right). The reference results correspond to the hyperparameters considered in the main experiments. All results are reported as the mean and standard deviation over three runs with different random seeds.

In Figure 17, we can find the corresponding results for the ablation analysis focusing on the cosine similarity of the learned Lie algebra basis to the ground truth instead of the performance metric. For the weight terms of the loss function, we can see that modifying $\alpha$ (the weight of $\mathcal{L}_{obj}$) has little effect on the symmetry discovery process. Modifying $\lambda$ and $\nu$ also has little impact below a threshold of 0.1, but when those terms are increased too much, they lead to a degraded symmetry discovery performance, as the difference of the augmentations from the original data ($\lambda$) or the sparsity of the learned basis ($\nu$) become too prominent in the loss. Furthermore, $\beta$ also has a relevant impact on the symmetry discovery process, where we observe that higher values of that weight help the performance of the model. Lastly, the number of augmentations per sample has little impact on the symmetry discovery process for values below 16, but at that point the performance degrades, possibly for the same reason that we hypothesized for the performance metric.

Next, we also explore the effect of the chosen range $\gamma$ for the uniform sampling of coefficients for the linear combination of elements of the Lie algebra basis, which is then mapped to a group element. This is a hyperparameter that we did not extensively tune in our experiments, and its values were chosen mainly seeking to encourage the sampling of diverse elements of the group while not being excessively large to avoid distortions in the uniformity of the sampling (e.g., for rotation groups the sampled elements would "wrap around" for larger ranges, thus distorting the effective sampling density of the group elements).

In order to verify our intuition and solidify our explanation in that regard, we run an additional ablation experiment for the $\gamma$ hyperparameter using the RotatedMNIST dataset, the results of which are shown in Table 23. In these results, we would start observing the "wrap around" effect starting at $\gamma = 5.0$ if the model were to learn the ground truth symmetry, which may explain the decrease in performance and cosine similarity for that and the larger values of this hyperparameter. In the case of $\gamma = 13.0$, we hypothesize that the slight increase in the performance and similarity of the basis is

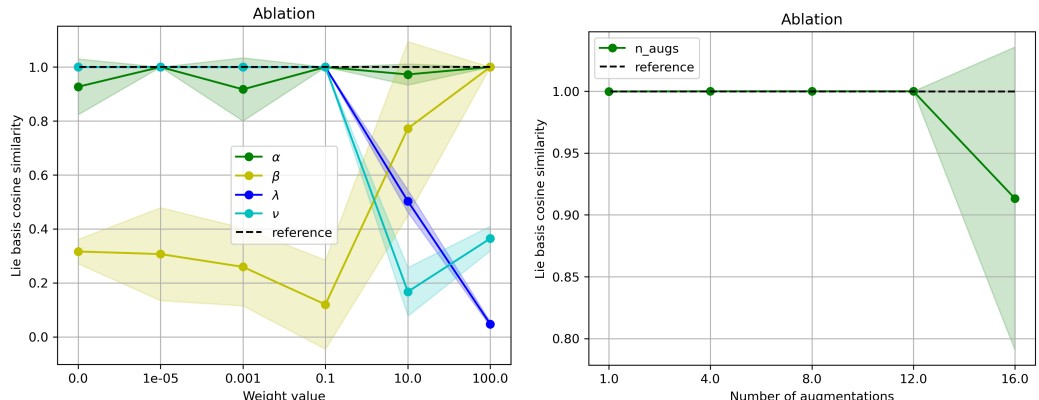

Figure 17: Ablation results for the cosine similarity of the learned Lie algebra basis to the ground truth in the RotatedMNIST dataset for varying values for the weight terms of the loss function (left) and the number of augmentations per sample (right). The reference results correspond to the hyperparameters considered in the main experiments. All results are reported as the mean and standard deviation over three runs with different random seeds.

due to the fact that learning the ground truth Lie algebra basis would lead to rotations approximately in the range $[-2\pi, 2\pi]$, thus circumventing the issue.

Table 23: Results for the ablation of $\gamma$. We report the mean and standard deviation for both metrics across three runs using different random seeds. The accuracies are expressed as percentages.

| $\gamma$ | Test accuracy | Cosine similarity Lie algebra basis |
|---|---|---|
| 1.0 | $97.73_{\pm 0.17}$ | $0.9999_{\pm 0.0001}$ |
| 2.0 | $98.04_{\pm 0.22}$ | $1.0000_{\pm 0.0000}$ |
| 3.0 | $99.08_{\pm 0.06}$ | $0.9997_{\pm 0.0002}$ |
| 5.0 | $95.97_{\pm 3.24}$ | $0.7860_{\pm 0.3026}$ |
| 8.0 | $93.11_{\pm 3.79}$ | $0.5706_{\pm 0.3571}$ |
| 9.0 | $94.93_{\pm 4.94}$ | $0.6783_{\pm 0.4549}$ |
| 10.0 | $92.56_{\pm 4.19}$ | $0.6273_{\pm 0.2656}$ |
| 11.0 | $89.28_{\pm 2.96}$ | $0.3709_{\pm 0.2551}$ |
| 12.0 | $96.74_{\pm 2.14}$ | $0.6881_{\pm 0.4381}$ |
| 13.0 | $96.98_{\pm 0.72}$ | $0.9812_{\pm 0.0178}$ |
| 14.0 | $93.32_{\pm 2.99}$ | $0.7236_{\pm 0.2095}$ |
| 15.0 | $91.89_{\pm 2.20}$ | $0.5346_{\pm 0.0607}$ |
| 16.0 | $93.52_{\pm 0.91}$ | $0.2104_{\pm 0.1597}$ |
| 17.0 | $90.28_{\pm 3.33}$ | $0.2779_{\pm 0.1739}$ |
| 21.0 | $89.00_{\pm 4.13}$ | $0.0006_{\pm 0.0006}$ |

Finally, we also explore the symmetry discovery performance of SEMoLA under a misspecified Lie algebra basis cardinality $C$ in Figure 18. As we can see, under the slight misspecification of setting $C = 2$, we observe redundancy across the different elements of the learned basis, which shows that our model is able to robustly discover the correct basis. In the more extreme cases of setting $C = 4$ or $C = 6$, we start observing a slight degradation in the discovered basis, which is somewhat noisier. In addition, while some elements in the basis are inverses of one another, their linear combination would still lead to a valid rotation augmentation. Therefore, while knowing the correct (or a close

approximation) of the basis cardinality a priori helps the symmetry discovery performance of the model, we have shown that it can be robust in scenarios where that knowledge may not be available.

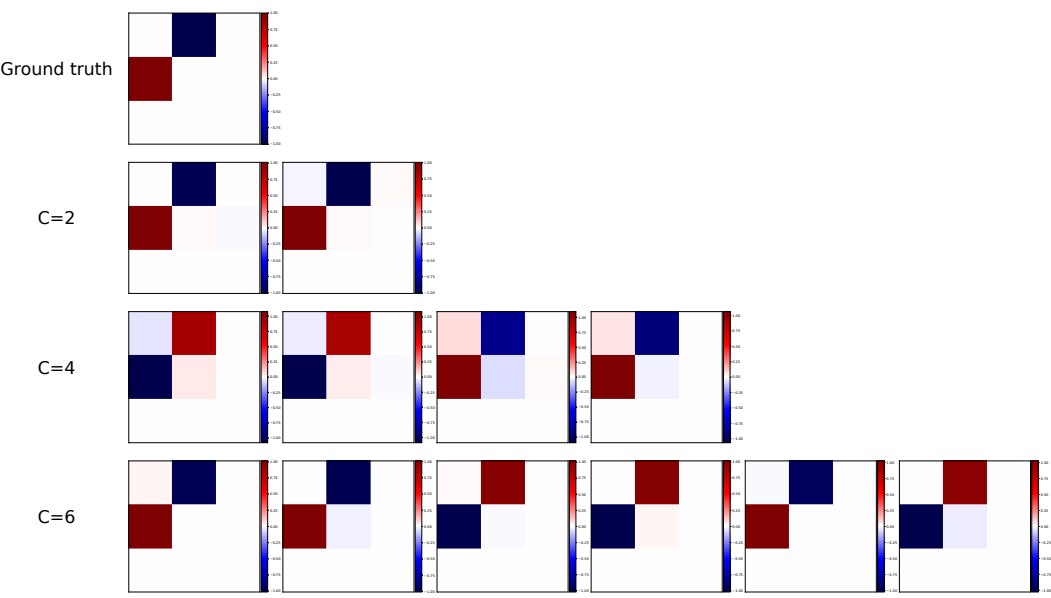

Figure 18: SEMoLA is able to correctly discover a representation of the underlying symmetry even when the cardinality of the Lie algebra basis is misspecified in the RotatedMNIST dataset.

### A.10 DATASET WITHOUT SYMMETRIES

As an additional experiment, we investigate the behavior of SEMoLA when applied to a dataset that lacks any symmetry in order to determine whether it learns an arbitrary spurious symmetry or if, on the contrary, it discovers a representation that can be easily used to diagnose the lack of symmetry in the dataset and that does not negatively impact the performance of the method.

In that sense, we define a synthetic dataset where the inputs $x \in \mathbb{R}^5$ and labels $y \in \mathbb{R}$ are sampled from uniform distributions. As a result, this dataset does not contain any symmetry in its map from input to output, and, ideally, a symmetry discovery method should learn a basis that corresponds to the 0 matrix, so that when sampling group elements, an identity transformation is applied, as no other similarly parameterized transformation should be strictly beneficial for the downstream performance. Nevertheless, SEMoLA's implementation enforces a constant norm on the Lie algebra basis during training, and, thus, that cannot occur. What we observe instead is that in this case the norm of the matrix concentrates in a single entry, which changes in every one of the runs with different random seeds that we executed, as we can see in Figure 19.

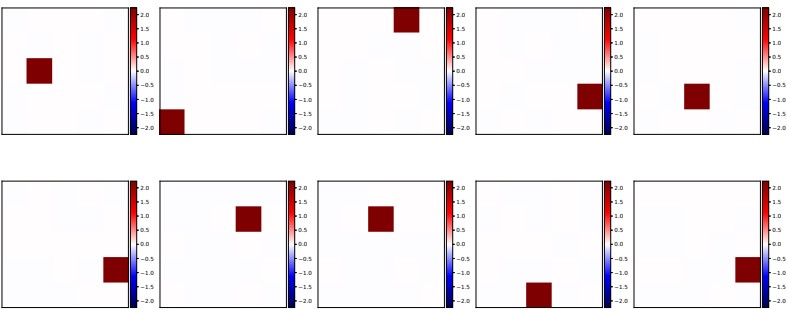

Figure 19: Lie algebras learned in each of 10 runs with different random seeds for the synthetic dataset without symmetries.

As a result, these observations hint that our method could potentially also be applied as a way of determining the extent of symmetry in a dataset, i.e., if the learned Lie algebra basis is similar across runs, the data potentially has a stronger symmetry with respect to the prediction task.

### A.11 QUANTUM FIELD THEORY AMPLITUDE PREDICTION

As an additional experiment, we study the problem of defining a neural surrogate for quantum field theoretical amplitudes (Spinner et al., 2024; Aylett-Bullock et al., 2021; Badger and Bullock, 2020), which is of great importance in the physics community due to it being central to the theoretical predictions that LHC measurements are compared to. In particular, these amplitudes describe the un-normalized probability of interactions of fundamental particles as a function of their four-momenta, which are exactly Lorentz-invariant. Following Spinner et al. (2024), we study the setting $q\bar{q} \to Z + ng$, i.e., the production of a Z boson with $n = 1, \ldots, 4$ additional gluons from a quark-antiquark pair. In particular, we focus on the scenario where $n = 2$ and we generate and split the dataset as in that work.

We train SEMoLA using the baseline Transformer model defined in Spinner et al. (2024) as the base prediction model, and with a learnable Lie algebra basis of cardinality $C = 6$ corresponding to that of the restricted Lorentz group $SO^+(1, 3)$, i.e., the connected identity component of the Lorentz group. From Figure 20, we can observe that SEMoLA is able to approximately learn the correct group. Even though some of the elements of the learned Lie algebra basis (especially those corresponding to boosts) are not exactly equivalent to their ground truth counterparts, we believe that this is more of a side-effect caused by the symmetries that exist in this particular dataset rather than a partial imprecision of our method.

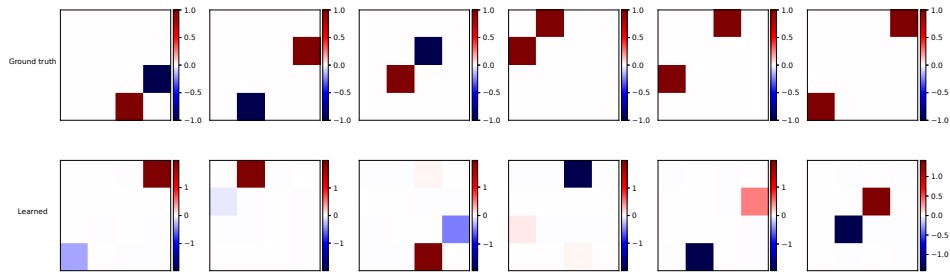

Figure 20: Comparison of the ground truth Lie algebra basis with the one learned by SEMoLA in the QFT amplitudes dataset.

This hypothesis is partly supported by the results in terms of the test error comparison with other baseline models (L-GATr (Spinner et al., 2024) and Transformers with and without data augmentation) that we include in Figure 21, which show that a Transformer trained with augmentations from the ground truth restricted Lorentz group (Transformer_aug) performs worse than the same Transformer without data augmentation. As a result, it is possible that the characteristics of this dataset are such that the samples do not necessarily reflect the action of the entire group, which could potentially explain both the higher accuracy of certain elements of the Lie algebra basis learned by SEMoLA, as well as the slightly negative effect of data augmentation.

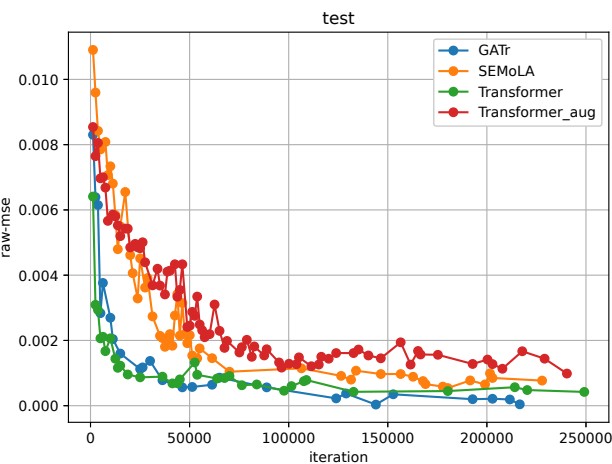

Figure 21: Comparison of the test MSE for different baseline models and SEMoLA in the QFT amplitudes dataset.

