# OpenReview forum: "Learning Equivariant Models by Discovering Symmetries with Learnable Augmentations"
_ICLR.cc/2026/Conference — Submitted to ICLR 2026_

### Official Review · Reviewer_VRqe · 2025-10-27

**Soundness:** 2
**Presentation:** 2
**Contribution:** 1
**Rating:** 2
**Confidence:** 4

**Summary:**

The authors propose SEMoLA, a method for unsupervised symmetry discovery consisting of two components: 1). An augmenter which learns a Lie algebra basis and samples augmentations which are applied to input data; and 2). A secondary network which takes in both original and augmented data together to make predictions. Experiments are performed on several datasets.

**Strengths:**

To my knowledge the composition of the augmenter with an unconstrained model is novel, and the authors provide extensive ablations in the supplement to justify their approach.

**Weaknesses:**

Unfortunately, there are several problems with this paper.

- First, the authors do not consider non-connected Lie groups and state that this is consistent with prior work. This is not true -- the authors   fail to cite or referece the following recent work:

  1). Neural Fourier Transform: A General Approach to Equivariant Representation Learning (Koyama et al, ICLR 2024) https://openreview.net/ forum?id=eOCvA8iwXH

  2). Neural Isometries: Taming Transformations for Equivariant ML (Mitchel et al, NeurIPS 2024) https://arxiv.org/pdf/2405.19296

  in which both of these models are shown to discover symmetries corresponding to non-connected and non-compact Lie groups (e.g. SL(3, R) and SL(2, C)).  Both of these methods appear to handle much more challenging cases than the proposed approach and go further in considering a variety of different applications, including some with real world data.

- To this point, the experiments are unconvincing. Two out of the three experiments in the main paper consider very simple, synthetic datasets (Rotated MNIST, and N-body dynamics) and the proposed method is outperformed by EMLP + LieGAN on the remaining dataset. Furthermore, the symmetry groups considered are very simple (SO(2), SE(3)) and are already well-studied in the symmetry discovery literature.

- Other outstanding problems include a poor description of the actual approach itself. Specifically, the mechanics of the proposed approach are unclear. For instance, how is there assumed to be a well defined action of $g$ on $\hat{y}$ as implied in Equation (2). Furthermore the loss in Equation (1) contains five terms which seem to require careful balancing, which calls in to question both  the robustness and soundness of the method.

Overall, the proposed method does not appear to move the field of symmetry discovery forward in a significant way, and so I do not recommend acceptance.

**Questions:**

How do you ensure that Lie algebra actually forms a basis, and is closed under the Lie bracket? If it is not closed, then it isn't really a true Lie algebra basis. I am aware that the LieGAN line of work also does not ensure this, but it seems like an important outstanding theoretical problem that needs to be addressed when learning generators.

---

> ### Author Response · Authors · 2025-11-20
> **Response to Reviewer VRqe (Part 1/2)**
>
> We would like to thank the reviewer for their thoughtful comments and for remarking on the novelty of our proposal and its justification through our extensive ablation experiments. We address the raised concerns below.
>
> **- Goal of the approach**
>
> We would like to start by briefly reemphasizing the main goal of our proposed approach, which is to introduce a paradigm for the definition of learnable equivariant models, i.e., models that learn to produce approximately equivariant predictions with respect to a priori unknown symmetries that they discover through end-to-end supervised training.
>
> **- Recent related works**
>
> We thank the reviewer for pointing out these interesting works, which we agree are related to our approach and that we have made sure to reference in the related work section of the revised version of our manuscript for completeness (see lines 94-100). However, even though they consider some Lie groups in the context of equivariant machine learning, they are not attempting to solve exactly the same problems that we are exploring in our work.
>
> In particular, 1) proposes an unsupervised method for learning the latent linear action of a potentially unknown group (although most of the experimental results assume knowledge of the group). As a result, this method is not designed for interpretable discovery of unknown symmetry groups or for the direct generation of approximately equivariant prediction, as SEMoLA is concerned with. Moreover, it requires that the dataset consists of tuples of observations that correspond to separate orbits of the group action, which is not an assumption that would generally hold in practice for arbitrary datasets and constitutes a significant limitation to the applicability of that approach that our model does not suffer from.
>
> Similarly, 2) focuses on the problem of self-supervised representation learning (conversely to our focus on supervised learning) and requires training an additional equivariant model on the learned latent space in order to actually produce equivariant predictions (contrary to our end-to-end single model approach). In addition, this method is not designed to discover an interpretable representation of an unknown symmetry group, and it also assumes the existence of a dataset of pairs of observations related by a group action, which again constitutes an important limitation and restriction to its applicability.
>
> **- Description of the approach**
>
> Firstly, regarding the assumption that there exists a well-defined action of the group on the model's predictions, it follows from the well-defined action on the ground truth labels. In particular, the training objective of our method, mainly through the equivariance loss term, encourages the model to generate outputs that are consistent with the learned group actions. We thank the reviewer for pointing out this potential ambiguity, and we have added a brief explanation to lines 256-259 of the revised manuscript seeking to clarify it.
>
> Secondly, we agree that the number of terms in the loss function can raise questions about the robustness of the approach. For that reason, we explored its sensitivity to different coefficient values in the ablation experiments included in Appendix A.9. The results from that analysis show that our model is very robust to the assigned values for those coefficients in terms of both task performance and symmetry discovery performance, especially if the values are in reasonable ranges, i.e., the regularization terms are not weighted excessively to overpower the two main loss terms corresponding to the empirical and equivariance losses. We have made sure to emphasize the existence of those ablation experiments, as well as the observed robustness of our method in lines 242-244 of the revised paper.
>
> We hope that these comments have helped clarify these aspects of the paper, and we would be happy to further discuss and revise any additional parts of the description of the method to increase their clarity.
>
> **- Lie bracket closure**
>
> Just like prior symmetry discovery works, such as LieGAN or Augerino+, we do not explicitly enforce closure of the learned Lie algebra basis under the Lie bracket. However, the minimization of the proposed loss function encourages that the sampled transformations behave as group actions, which, combined with the way the sampling of those transformations is parameterized, favors the convergence towards a Lie algebra basis. The reasons behind not enforcing closure under the Lie bracket are mainly related to the associated computational complexity and increased difficulty that the optimization problem would require. Just like in prior symmetry discovery methods, we believe that this reparameterization choice is well-justified by those reasons, and it is also supported by the success of our experimental results.

---

> > ### Author Response · Authors · 2025-11-20
> > **Response to Reviewer VRqe (Part 2/2)**
> >
> > **- Experimental evaluation**
> >
> > We agree that some of the datasets that we study in our experiments, e.g., RotatedMNIST, can be considered to be quite simple.
> > However, we believe that the possible apparent simplicity of some of the datasets does not necessarily imply the simplicity of the associated task we are trying to solve with them, i.e., joint symmetry discovery and equivariant predictions.
> >
> > For instance, already for RotatedMNIST we can see many of the baselines struggle to either discover symmetries or produce correct predictions in the out-of-distribution version of the dataset, while our proposal performs robustly while addressing both tasks jointly instead of individually as the baselines. In the case of the N-body dynamics dataset, we can see how even in the in-distribution setting, the baseline symmetry discovery methods fail to learn the correct symmetry unless they are provided with significant priors, which highlights the inherent difficulty associated with this task. While it is true that EMLP + LieGAN provides a slightly higher performance than SEMoLA, that method corresponds to the separate training of a symmetry discovery method (LieGAN) and the subsequent definition and training of an EMLP based on the discovered symmetry, while our method jointly performs both tasks in a single end-to-end fashion and with a simpler and less constrained model, which results in a considerably reduced training time (see, for instance, Table 10).
> >
> > Additionally, we also study other more challenging datasets throughout the paper and the appendix, which we believe help validate the good performance of our proposal.
> > For instance, our experiments with the QM9 dataset (see Section 5.3 and Appendix A.6) constitute a more complex and realistic scenario, with, most notably, a more complex symmetry discovery scenario with a higher cardinality of the Lie algebra basis, which our model is able to very accurately recover. We also study the challenging problem of colorectal cancer detection with real-world data in Appendix A.7, which also shows the good performance and benefits of our approach with a wider variety of more complex models, such as ViT and EfficientNet. Moreover, we analyze the suitability of our method to discrete symmetries in Appendix A.8, which demonstrates the wider scope of its applicability.
> >
> > Regarding the symmetry groups that we consider, we focus on some that are indeed common (which facilitates the comparison with other approaches), but that are also quite useful and predominant in practice, especially in the design of equivariant models for scientific domains. Thus, they constitute a good proof of concept for our goal of showing that an unconstrained model can be made equivariant with respect to an a priori unknown symmetry through an end-to-end method that jointly performs symmetry discovery and softly enforces the base model to produce approximately equivariant predictions. In that sense, we believe that the extensive experimental results that we include throughout both the main body of the paper and the various appendices are sufficient to verify the success of our proposal in tackling that very challenging and hitherto unresolved scenario.

---

> > > ### Comment · Reviewer_VRqe · 2025-11-26
> > >
> > > Thank you for your detailed comments.
> > >
> > > Could the authors comment on how the proposed model might be applied to handle data where the group action is not linear but protective (e.g. SL(3, R) acting on the plane or SL(2, C) acting on the Riemman sphere by fractional linear transformations).
> > >
> > > Furthermore, the lack of surjectivity of the exponential map for certain groups is a known problem. However, there exists modifications of the exponential map (sometimes called the Riemannian exponential) that are in fact surjective. For instance, see Equation 8.2 of [1].  It appears this map could potentially replace the standard exponential in the proposed framework, opening the door for more complex symmetries.
> > >
> > > However, most real world data (e.g. in vision) does not behave nicely and transformations are often ill-described by group actions. It seems that neither the exponential map nor Lie Groups themselves are actually fundamental to the proposed approach. Can the authors provide some thoughts about how one might replace the exponential and Lie algebra with a learnable, black-box augmentation generator?
> > >
> > > [1] Left-Invariant Riemannian Geodesics on Spatial Transformation Groups, Zacur et al. 2014.
> > > https://epubs.siam.org/doi/pdf/10.1137/130928352?casa_token=AGInibhBYo8AAAAA:fNcel7a7uFEhgdJsWoua9_l6dPWNV1HKeAHKZKW5KNM7HTfQOdk_WYrlkN4xwhYJsdv2TSQ4

---

> > > > ### Author Response · Authors · 2025-12-03
> > > > **Response to Reviewer VRqe (Part 1/2)**
> > > >
> > > > We thank the reviewer for their response and suggestions. We answer the newly posed questions below.
> > > >
> > > > **- Projective group actions**
> > > >
> > > > Our current implementation assumes that the group acts linearly on inputs and outputs via a representation, which is in line with many symmetry-discovery and equivariant learning methods and makes the learned symmetries easy to interpret and compare. This is a modeling choice rather than a fundamental limitation of the proposed framework.
> > > >
> > > > In the projective cases mentioned by the reviewer (e.g., SL(3,R) acting on the plane, SL(2,C) acting on the Riemann sphere), one could introduce an embedding that lifts the inputs to homogeneous coordinates, where the action of the group becomes linear, apply our augmentation method in that lifted space, and then project back to the original domain. This would allow the model to handle projective group actions with only additional pre- and post-processing layers, while leaving the core symmetry-learning mechanism unchanged.
> > > >
> > > > Extending the architecture to treat more general non-linear group actions without such a linearization step would require additional design and analysis, which we leave for future work, but we view this as a natural next step building on the present linear case.
> > > >
> > > > We hope these remarks clarify how our method could be adapted to projective and more general non-linear group actions.
> > > >
> > > > **- Riemannian exponential**
> > > >
> > > > We thank the reviewer for the useful reference [1] and suggestion.
> > > >
> > > > Indeed, following the Hopf-Rinow theorem, the Riemannian exponential is surjective for manifolds that are connected and complete as metric spaces.
> > > > In other words, while the Riemannian exponential can be surjective on certain manifolds, this is not guaranteed in general, and even when it is surjective, it does not preserve the algebraic properties that we rely on.
> > > > In addition, if we specifically consider matrix Lie groups, it can be shown that the Riemannian exponential when equipped with an algebra at the identity agrees with both the Lie exponential and the matrix exponential whenever the metric is bi-invariant (see Appendix C in [A]).
> > > >
> > > > Moreover, we would like to point out that [1] shows that closed-form solutions of the Riemannian exponential can only be defined for some groups and under particular invariant metrics.
> > > > Hence, such an exponential map is not well-suited for symmetry discovery in our setting, since it would introduce a circular dependency between the learned group and the prior knowledge of its geometry used for the definition of the exponential.
> > > >
> > > > We would like to thank the reviewer for encouraging us to explore alternative parameterizations of symmetries with different properties, and we agree that this is a very valuable direction to explore. However, since our setting relies on Lie groups and Lie algebras, a careful design and systematic analysis of such alternatives are required, and, therefore, we leave them as future work.
> > > >
> > > > **References:**
> > > >
> > > > [1] Left-Invariant Riemannian Geodesics on Spatial Transformation Groups, Zacur et al. 2014.
> > > >
> > > > [A] Lezcano-Casado, Mario, and David Martınez-Rubio. "Cheap orthogonal constraints in neural networks: A simple parametrization of the orthogonal and unitary group." In International Conference on Machine Learning, pp. 3794-3803. PMLR, 2019.

---

> > > > > ### Author Response · Authors · 2025-12-03
> > > > > **Response to Reviewer VRqe (Part 2/2)**
> > > > >
> > > > > **- Black-box learnable data augmentations**
> > > > >
> > > > > Thank you for this insightful comment and question. We agree that many real-world transformations, especially in vision, are not well described by exact group actions, and that a useful learnable augmentation mechanism does not have to be parameterized as a Lie-group action.
> > > > >
> > > > > In our framework, the role of the exponential map and Lie algebra is specifically to define an interpretable differentiable family of transformations acting on the data. In principle, this component could be replaced by a black-box augmentation generator $T_\theta(x)$, e.g., a neural network that takes an input x and outputs an augmented sample, while keeping the same objective that encourages the prediction model to be invariant/equivariant to $T_\theta$ (for example, cropping or color jittering in image classification tasks). However, designing such transformations without knowing the symmetries in advance would be challenging.
> > > > > One could optionally add soft regularizers to $T_\theta$ to promote group-like behaviour (approximate identity, invertibility, or composition), without committing to a specific Lie-group structure, but it would require careful design.
> > > > > In practice, enforcing meaningful compositional behavior and stability for a black-box method may require reintroducing some form of algebraic or geometric structure, bringing us back close to the spirit of Lie-theoretic models.
> > > > >
> > > > > Related black-box augmentation generators have already been explored in specific domains, such as anomaly detection for time series and self-supervised GNNs, as we discuss in the "Learnable augmentations" paragraph of the related work section (lines 128-138). These works demonstrate the effectiveness of such flexible augmentations but do not provide interpretable descriptions of the learned transformations. In contrast, our focus in this work is on a constrained parametrization that yields explicit generators and thus interpretable symmetries of the data, in addition to performance gains. Exploring a less constrained, black-box version of our augmentation module, building on the same training principle, would be an interesting direction for future work.

---

### Official Review · Reviewer_UrkV · 2025-11-01

**Soundness:** 3
**Presentation:** 3
**Contribution:** 2
**Rating:** 2
**Confidence:** 4

**Summary:**

This paper proposes a new method that takes an unconstrained base model and finds the symmetry and its extent underlying a dataset by end-to-end joint learning of the base model and group representations. This is done by restricting the scope to connected Lie groups with surjective Lie exponential, and learning the basis of the Lie algebra, such that data augmentations are produced from uniformly distributed coefficients under the basis. While this approach is similar to Augerino, a main difference is that the method does not assume fixed basis directions. The authors demonstrate that the proposed method outperforms Augerino+ and LieGAN and is competitive with the ground truth data augmentation in MNIST under SO(2), 2-body dynamics under SO(2), and QM9 under SE(3), including the cases where the training set contains only a restricted set of augmentations.

**Strengths:**

S1. The paper tackles the challenging problem of jointly discovering symmetry and its extent from data in the form of data augmentation, which is jointly used with an unconstrained model to produce (approximately) equivariant predictions.

S2. The empirical results show that the proposed method outperforms Augerino and LieGAN in the three experimented setups.

**Weaknesses:**

W1. A main weakness of the work is that the experimented setups only consider Lie groups with low-dimensional Lie algebras and with numerically well-behaved generators (SO(2), SE(3), and small permutation groups), such that it is hard to verify whether the method can indeed discover symmetries in nontrivially hard problem instances, e.g., discovering affine transformations and/or homographies from transformed MNIST images [1], and/or discovering Lorentz transformations from jet tagging as in the LieGAN paper.

W2. It was unclear to me whether, for data dimension $d$ (e.g., $H\times W$ for monochrome images), the group action is assumed to be unknown linear maps on $\mathbb{R}^d$, or is assumed to act homogeneously on coordinates in $\mathbb{R}^2$ or $\mathbb{R}^3$. If it is the latter, then I believe the algorithm has access to a nontrivial amount of knowledge about data transformations prior to the learning (specifically, low-dimensionality and factorization structure of the action), and it is unclear whether we have similar amounts of information in practical setups when symmetry is unknown.

W2. Line 154-155: As far as I know, it is not true for every connected (matrix) Lie group that any element can be written as the output of Lie exponential. In case of noncompact groups the exponential map can be nonsurjective. A counterexample can be found in [2]. This can be a limiting factor of the applicability of the method.

[1] MacDonald et al., Enabling Equivariance for Arbitrary Lie Groups, CVPR 2022.

[2] https://math.stackexchange.com/questions/348699/showing-that-the-exponential-map-mathrmexp-mathfraksl2-mathbbr-to-ma

**Questions:**

I have no particular questions but would like to hear the authors' response to the weaknesses.

---

> ### Author Response · Authors · 2025-11-20
> **Response to Reviewer UrkV**
>
> We would like to thank the reviewer for the insightful review and helpful comments, as well as for highlighting both the challenging nature of the problem our method addresses and its strong empirical performance in comparison to the considered baselines. We answer the remaining concerns below.
>
> **- Complexity of the considered Lie groups**
>
> Regarding the symmetry groups that we consider, we focus on some that are indeed common (which facilitates the comparison with other approaches) and have low-dimensional Lie algebras and well-behaved generators, but that are still quite useful and prevalent in practice. Thus, we believe they constitute a good proof of concept for our goal of showing that an unconstrained model can be made equivariant with respect to an a priori unknown symmetry through an end-to-end method that jointly performs symmetry discovery and softly enforces the base model to produce approximately equivariant predictions. In that sense, we believe that the extensive experimental results that we include throughout both the main body of the paper and the various appendices are sufficient to verify the success of our proposal in tackling that very challenging and hitherto unresolved scenario.
>
> We agree that some of the datasets that we study in our experiments, e.g., RotatedMNIST, can be considered to be quite simple (although other symmetry discovery methods already struggle in the out-of-distribution version of this dataset).
> However, we believe that the QM9 and the 2-body dynamics already pose nontrivially hard scenarios for the joint task of symmetry discovery and generating approximately equivariant predictions. This is particularly evident due to the inability of the considered baselines to discover the correct symmetry without auxiliary knowledge in the 2-body dynamics experiments, as we show in Section 5.2 and Appendix A.5.
>
> Nevertheless, for the sake of completeness, we have performed an additional experiment, which we include in Appendix A.11, to study the ability of our model to discover symmetries corresponding to the Lorentz group. As the results show, SEMoLA is able to approximately discover such a symmetry and it provides performance comparable to the hard equivariant baseline L-GATr. Given the time required for setting up this experiment and the amount of resources needed for carrying it out, these results correspond to a somewhat limited finetuning and exploration of the performance of our method. Thus, we believe that the quality of the obtained results, despite those limitations, is a good indication of the potential of SEMoLA to be readily applied to this type of challenging problem.
>
> **- Group action**
>
> In this regard, we follow the approach of other symmetry discovery methods, such as LieGAN and Augerino+, and apply the group action differently depending on the considered data type. For image data, i.e., RotatedMNIST (Section 5.1 and Appendix A.4) and colorectal cancer (CRC) detection datasets (Appendix A.7), we assume the group action is applied on the pixel coordinates, while for all other data types in our experiments, the group action is applied over the data dimension $d$. Even though it is true that in the case of image data, this design choice introduces a significant prior into the type of transformations that our approach is trying to learn, we believe that it still corresponds to useful and interesting scenarios, where there exists some prior knowledge about the structure of the group action but not about its corresponding group.
> We have made sure to emphasize this difference in the symmetry discovery setting for image data in lines 317-320 for RotatedMNIST and 1356-1360 for CRC.
>
> **- Surjectiveness of the exponential map**
>
> This is a very pertinent observation for which we would like to thank the reviewer. We have remedied this imprecision through a modification to lines 164-165 of the revised manuscript. While exponential maps can generate the elements of a connected Lie group, they are indeed not necessarily surjective for noncompact groups. This nuance can slightly restrict the applicability of our approach, but it is a common limitation that is shared by the most popular alternative symmetry discovery methods, e.g., LieGAN and Augerino+. They follow the same path, to improve efficiency and simplicity of the reparameterization of the distribution over Lie group elements, while still allowing application to a wide variety of important and common symmetries found in many datasets and tasks, such as SO(n) or SE(n).

---

> > ### Comment · Reviewer_UrkV · 2025-11-25
> >
> > Thank you for the thoughtful response. Regarding the added experiment in Appendix A.11, can the authors elaborate a bit on why LieGAN seems to be able to discover the Lie algebra basis of SO(1, 3)+ (see Figure 5 therein), while the proposed method does only approximately in Figure 20? I am not entirely convinced by the explanation provided in Appendix A.11 (Lines 1800-1824), because if it is true there is no reason LieGAN has to succeed.
> >
> > For the second item (group action), can the authors provide specific examples of applications or problem scenarios where one has a nontrivial knowledge of group actions, while not knowing the underlying groups? Adding this could serve as a reasonable justification of the experimental setups.
> >
> > For the last item (exponential), it may be beneficial to explicitly write out in Lines 162-167 what are the classes of groups that can be discovered in principle by the method (and prior methods). As a related question, what would happen if the method (or prior methods) attempts to discover symmetry from a dataset of which ground truth symmetry has a non-surjective exponential map?

---

> > > ### Author Response · Authors · 2025-12-03
> > > **Response to Reviewer UrkV (Part 1/2)**
> > >
> > > We thank the reviewer for their reply and additional suggestions. We address the remaining questions below.
> > >
> > > **- Lorentz group symmetry discovery (LieGAN vs SEMoLA)**
> > >
> > > We thank the reviewer for raising this question and encouraging us to compare the experimental setups in more detail in order to clarify the source of the discrepancy between SEMoLA's and LieGAN's symmetry-discovery performance for the Lorentz group.
> > >
> > > The main difference arises from the fact that the dataset we use does not coincide with that of the original LieGAN paper. In their implementation, LieGAN is trained on one dataset for symmetry discovery, while the downstream prediction models use a differently processed version of the data. In contrast, our approach is trained end-to-end for prediction. Hence, we decided to work with a different dataset, which enables a coherent analysis within our framework but prevents an exact comparison with the results reported in the original LieGAN paper.
> > >
> > > In addition, the LieGAN implementation for this experiment uses a group cardinality of 7, whereas the restricted Lorentz group has cardinality 6. This decision effectively relaxes the constraint on the learned basis, since one of the learned generators does not correspond to the target symmetry. When we retrain LieGAN with the true cardinality of 6 and keep the hyperparameters specified by the authors, the resulting Lie algebra basis is considerably noisier and less accurate than the one reported in their paper. This suggests that the reported performance is sensitive to the hyperparameter configuration. On the other hand, throughout our experiments (including the Lorentz-group experiment on our dataset), we perform minimal hyperparameter tuning and show in Appendix A.9 that SEMoLA is robust to these choices.
> > >
> > > Moreover, note that Table 3 of the LieGAN paper shows that all prediction models achieve very similar performance on their Lorentz-group dataset, indicating that this benchmark is relatively simple and close to saturated. To evaluate symmetry discovery in a more challenging setting, and to address the request for an example involving a more complex symmetry group, we therefore consider the task of estimating Quantum Field Theory amplitudes. In this setting, the results in [A] show that a Transformer can attain performance comparable to an explicitly equivariant architecture, which raises the question of whether full Lorentz equivariance is actually required. Our experiments support the hypothesis that only a subset of generators is important for this dataset. A systematic investigation of this hypothesis lies beyond the scope of the present work, but we believe it constitutes an interesting direction for future study.
> > >
> > > We hope that these additional experimental details and clarifications help better interpret the differences between LieGAN's and SEMoLA's symmetry-discovery behavior in the context of the Lorentz group.
> > >
> > > **References:**
> > >
> > > [A] Spinner, Jonas, Victor Bresó, Pim De Haan, Tilman Plehn, Jesse Thaler, and Johann Brehmer. "Lorentz-equivariant geometric algebra transformers for high-energy physics." Advances in neural information processing systems 37 (2024): 22178-22205.

---

> > > > ### Author Response · Authors · 2025-12-03
> > > > **Response to Reviewer UrkV (Part 2/2)**
> > > >
> > > > **- Nontrivial knowledge of group actions with no knowledge of the group**
> > > >
> > > > An example of a domain in which one can have a conceptual knowledge of the group actions but not necessarily of the exact group itself is medical imaging. In that field, one highly desirable property is invariance to diffeomorphic transformations that preserve the topological properties of the data, since anatomical structures need to be aligned precisely without grid folding to enable accurate analyses and diagnoses (see, for instance, [B]). Particular dataset instances in this domain, however, do not necessarily reflect the action of the entire diffeomorphism group, since that could cause different anatomical shapes to be treated as identical.
> > > > Therefore, while in this problem setting there is some prior knowledge about the type of transformations that we would like a model to be invariant to as well as how they are expected to act on the data, knowledge of the exact transformations, in this case the specific subgroup of the diffeomorphism group, may not be available a priori and would be useful to learn.
> > > >
> > > > Other examples can be found in the field of robotics, for instance, when learning control policies of unmanned aerial vehicles [C]. In such scenarios, we know a priori that the prediction model should be equivariant to rigid body transformations, e.g., SE(3). However, not the entire group is expected to act on the data, with, for instance, rotations being restricted to the axis corresponding to the gravity vector. Nevertheless, the specific subgroup of SE(3) that the data should be equivariant to could depend on the environment the robot is deployed in, as well as its specific sensor calibration. Thus, in this case, we also have considerable prior knowledge on the type of the group and its action, but the specific group can be problem-dependent, and discovering it would be highly beneficial since it would simplify the design of problem-specific equivariant prediction models.
> > > >
> > > > Our experimental setups for image data are meant as controlled instances of precisely this situation: the type of symmetry and its action are known at a high level, but the exact group (or subgroup) is not specified and is instead discovered from the data. These examples, therefore, provide concrete motivation for the kind of partial-prior, group-discovery setting we study.
> > > >
> > > > **- Exponential map and non-surjectiveness**
> > > >
> > > > We thank the reviewer for their suggestion, and we agree that including a detailed discussion of the groups that can be discovered by our method is important to properly understand the scope of our paper.
> > > >
> > > > In the revised manuscript (lines 164-170), we have provided such a description, where we emphasize that our method is in line with existing continuous symmetry discovery approaches, which also adopt the same reparameterization of a Lie group through its associated Lie algebra. That parameterization implies that our method can only fully describe Lie groups that are connected and compact, since the exponential map would be surjective in such cases.
> > > >
> > > > Regarding scenarios in which the group truth symmetry has a non-surjective exponential, the Lie algebra learned through this parameterization would correspond to a single connected component of the group. We thank the reviewer for raising this interesting question, and we have also included this information in the same revision of the manuscript.
> > > >
> > > > Extending our method to be applicable to a wider range of groups while trying to minimize the associated increase in computational complexity would be an interesting direction to explore in future work.
> > > >
> > > > **References:**
> > > >
> > > > [B] Matinkia, Mohammadjavad, and Nilanjan Ray. "Learning diffeomorphism for image registration with time-continuous networks using semigroup regularization." arXiv preprint arXiv:2405.18684 (2024).
> > > >
> > > > [C] Yu, Beomyeol, and Taeyoung Lee. "Equivariant reinforcement learning for quadrotor UAV." arXiv preprint arXiv:2206.01233 (2022).

---

### Official Review · Reviewer_Sgvd · 2025-11-01

**Soundness:** 4
**Presentation:** 3
**Contribution:** 3
**Rating:** 6
**Confidence:** 3

**Summary:**

This paper proposes a method to discover symmetries from data by learning augmentations. Specifically, the first module, LieAugmenter, learns a Lie algebra basis from the data. Lie group elements are then sampled and applied to the original data. The second module then takes in the augmented inputs and learns a task-specific function. The method uses several regularization terms to learn the correct symmetry. In experiments on RotMNIST, N-body dynamics, and QM9, SEMoLA outperforms other baselines such as LieGAN or Augerino.

Overall, this paper provides a good contribution to the area of symmetry discovery, beating current baselines.

**Strengths:**

- The method doesn't seem to rely on the distribution of Lie algebra basis coefficients and can use a uniform distribution.
- SEMoLA forgoes adversarial training leading to more stability and can be used end-to-end with the task function
- There are extensive experiments on various datasets and scenarios, including the experiments in the appendix.

**Weaknesses:**

- One small weakness is that this method relies on connected Lie groups. However, this is a common assumption taken in many other symmetry discovery papers as well.
- There are numerous regularization terms to balance. How sensitive is this method w.r.t. to the regularization coefficients?
- The training times seem somewhat longer than other baselines. Is this because the model needs to train on K augmented samples? Is the method sensitive to the value of K?
- Doesn't Augerino also learn a Lie algebra basis as done in SEMoLA? Or is it only learning ranges of augmentations?
- I believe L-conv (Dehmamy et al. 2021) also learns a Lie algebra basis in the conv layer. Does decoupling the learning of the basis with the task as done here (LieAugmenter + unconstrained model) compared to learning them together as in L-conv have a big impact on performance?
- In order to learn discrete groups (as in Appendix A.8.1) was it necessary to modify the distribution of the Lie algebra coefficients?

**Questions:**

See questions.

---

> ### Author Response · Authors · 2025-11-20
> **Response to Reviewer Sgvd (Part 1/2)**
>
> We would like to thank the reviewer for their encouraging feedback and for highlighting some of the advantages of our method with respect to existing symmetry discovery, as well as the extensiveness of our experimental evaluation. We address the remaining questions below.
>
> **- Robustness to regularization coefficients**
>
> We agree with the reviewer that given the amount of regularization terms in our proposed loss function, it is important to explore the sensitivity of our method to different values of their weight coefficients. In that sense, our results in Appendix A.9 show that our model is very robust to the assigned values for those coefficients in terms of both task performance and symmetry discovery performance, especially if the values are in a reasonable range, i.e., they do not overpower the main loss terms corresponding to the empirical and equivariance losses. We have made sure to emphasize the existence of those ablation experiments, as well as the observed robustness of our method in lines 242-244 of the revised paper.
>
> **- Training time**
>
> When compared to the base model for a particular experiment, our method is indeed expected to have a longer training time, mainly due to the generation and consideration of additional augmented samples. Similarly, when compared to the base model with ground truth augmentation, SEMoLA will require slightly longer training time due to the additional training parameters corresponding to the learned Lie algebra. However, when compared to hard equivariant models, SEMoLA will normally provide a considerably shorter training time due to its less constrained nature, as we can observe, for instance, in Tables 10 and 13 for the EMLP baselines. Moreover, we would like to emphasize that even though it is not captured by our training time comparisons (which correspond only to the training time for the prediction models), the total training time would be considerably higher as well for the models that are trained with LieGAN augmentations, since, as we highlight throughout the paper, they require a 2-step process where a LieGAN model is trained first and then augmentations from its learned symmetry are used when training an additional prediction model. We have made sure to emphasize this nuance in lines 736-742 of the revised manuscript. Lastly, we should mention that our current implementation could potentially be optimized further in order to reduce the associated time overhead, but our goal with this paper is to showcase the feasibility and success of our proposed approach, not to provide its optimal implementation.
>
> Regarding the potential impact of the number of augmentations on the performance of our method, we agree that this is also an important factor to explore. For that reason, we include an analysis of the sensitivity of our method to the number of augmentations in Appendix A.9. Our results indicate that SEMoLA is quite robust in terms of both its task performance and its symmetry discovery performance, and hint at a potential connection between the optimal number of augmentations and the batch size. We have highlighted the existence of those ablation experiments, as well as the observed robustness of our method in lines 210-211 of the revised manuscript.
>
> **- Differences with Augerino**
>
> Augerino is indeed designed to only learn ranges of augmentations, i.e., a distribution over the coefficients associated with the generators of a particular symmetry group, which is represented as a fixed Lie algebra basis. Conversely, Augerino+ corresponds to a relaxation of that method where the Lie algebra basis is learnable, as is the case for SEMoLA. However, already in the LieGAN paper, which proposed that modification, Augerino+ is shown to not be directly applicable for symmetry discovery, and its performance as a baseline in our experiments confirms its poor performance in general, which contributes to motivating the necessity of our approach.
> We have slightly rephrased our explanation of these methods throughout the 'Symmetry discovery' paragraph of Section 2 (i.e., lines 101-126), seeking to make these differences more evident.

---

> > ### Author Response · Authors · 2025-11-20
> > **Response to Reviewer Sgvd (Part 2/2)**
> >
> > **- Differences with L-conv**
> >
> > We believe that the main impact on performance would be due to the higher flexibility and adaptability of our approach. Firstly, in the sense that L-conv imposes significant hard constraints in the architecture, while our approach is applicable to any base model. As a result, SEMoLA can provide an increased performance in some cases simply due to the consideration of a more fitting or expressive architecture for the task being addressed. In addition, the decoupling and soft enforcement of equivariance through our multi-task loss objective allows the model to jointly consider the trade-off between equivariance and empirical loss, which allows it to be better suited for tasks with inexact and approximate symmetries. Lastly, L-conv can suffer from higher computational complexity due, for instance, to the repeated computation of matrix exponentials in every layer, while for SEMoLA they are only computed once when generating the augmentations.
> > We have made sure to emphasize these important differences in lines 113-118 of the revised paper for additional clarity.
> >
> > **- Distribution of Lie algebra coefficients for discrete groups**
> >
> > This is indeed an important consideration for the flexibility and robustness of our approach when knowledge of the discreteness of the group might not be available.
> > In our experimental analysis in Appendix A.8, we have observed that it is not strictly necessary to modify the distribution of the Lie algebra coefficients, but it does help to slightly improve the performance of our method in the experiments with discrete groups. In particular, that conclusion is supported by a comparison of the results for the modified distribution, denoted 'SEMoLA (discrete)', and the original one, which coincides with the one we apply in the other experiments, denoted 'SEMoLA (continuous)'.
> > We have highlighted these observations in lines 1452-1459 of the revised manuscript.

---

### Official Review · Reviewer_X127 · 2025-11-05

**Soundness:** 3
**Presentation:** 3
**Contribution:** 2
**Rating:** 6
**Confidence:** 4

**Summary:**

This work introduces a novel framework (SEMoLA) for discovering continuous symmetries directly from data and leveraging them as learned data augmentations during both training and testing. Instead of assuming a prior distribution over such group transformations, the proposed framework learns a Lie algebra basis whose exponential maps can be used to sample data augmentation applied to the model's input and output. The authors propose to jointly optimize for both the performance on the downstream tasks and the learned symmetry through a multi-task objective.  SEMoLA is empirically evaluated across multiple domains, showcasing how the proposed method can benefit the task's performance, while also providing interpretability regarding the symmetries of the given dataset and tasks.

**Strengths:**

- The proposed framework is agnostic to the choice of the network architecture and can be easily incorporated in a large range of possible tasks and models.
- The authors conduct a comprehensive evaluation of the different desing choices, including detailed ablation studies that assess the contribution of each component.
- The ability of the framework to provide interpretable learned augmentations can also be a useful tool in tasks where the main goal is to analyze or verify underlying symmetries, rather than solely optimize the performance of a downstream task.

**Weaknesses:**

- By fixing $\rho_y(g)=g$ or $\rho_y(g)=I$, the authors limit the framework to only learn the general group acting on the network's output rather than the representation acting on it. This assumption limits the applicability of the method in settings where the relevant representation is not obvious, which is common in many cases where this method could be impactful, since the appropriate transformation is unknown a priori
- The experiments focus mainly on datasets with useful and interpretable symmetries, where discovering the underlying transformation naturally benefits the task. However, the paper does not explore cases where such a transformation can be detrimental to the task, and the imposed regularizers that promote non-trivial symmetry discovery can harm the task.

**Questions:**

- Applicability without known representations: Is the method applicable in cases where the representation acting on the output or input is not known a priori? For example, in a typical image encoder, where we do not know if the output feature map should be interpreted as a scalar map or an equivariant vector field, how would SEMoLA handle such ambiguity?
- Behavior in the absence of true symmetries: What happens when no useful transformations are present in the data? Could the framework discover spurious symmetries that negatively affect generalization or stability?

---

> ### Author Response · Authors · 2025-11-20
> **Response to Reviewer X127**
>
> We would like to thank the reviewer for their thoughtful review and interesting suggestions, as well as for remarking on the wide applicability of our proposed method and the comprehensive nature of its evaluation. We answer the remaining questions below.
>
> **- Applicability without known representations**
>
> This is an interesting question because even though our current implementation does assume that the group representation acts linearly on inputs and outputs, we do not believe that to be necessarily a restriction on the applicability of our method, but rather merely a design decision that suits common scenarios in practice and aims to facilitate the clarity, interpretability, and comparison of the learned symmetries as is usually done in other symmetry discovery methods in the literature. Therefore, we believe that verifying the ability of our method to jointly discover both groups and their representations would be a promising future research direction that is worth exploring now that we have established its ability to learn symmetries with fixed group representations.
>
> Regarding the example of an image encoder, SEMoLA is designed to work in an end-to-end fashion, where the knowledge of whether the output is, e.g., an invariant scalar or an equivariant vector field, would potentially be known a priori, since it is defined by the task that we are trying to solve. Addressing the more complex scenario of learning equivariance to intermediate layers or encoders is a complex and interesting problem, but one not in the scope of this paper.
>
> **- Behavior in the absence of true symmetries**
>
> Our method is designed with the main motivation of increasing the flexibility of equivariant models, which are normally applied to data where the existence of some (potentially unknown or inexact) symmetry is usually expected. However, we agree that this is a very interesting scenario to explore in more detail, and, to that effect, we include new results in Appendix A.10.
>
> In particular, we define a new synthetic dataset consisting of random inputs and outputs in order to guarantee the lack of any symmetries. Ideally, we would expect a symmetry discovery method to learn a 0 matrix as the Lie algebra basis, so that when sampling group elements, an identity transformation is applied. However, since our method enforces a constant norm on the Lie algebra basis, that cannot happen. What we observe instead is that in this case the norm of the matrix concentrates in a single entry, which changes in every one of the runs with different random seeds that we executed.
>
> As the reviewer suggested, these observations indeed hint that our method could potentially also be applied as a way of analyzing the extent of symmetry in a dataset, i.e., if the learned Lie algebra basis is similar across runs, the data potentially has a stronger symmetry. We have made sure to emphasize this additional potential capability of our approach in our analysis of these results in Appendix A.10.
>
> In addition, since the loss that we propose is a joint minimization of the empirical and equivariance losses, our model is incentivized to learn symmetries that help the minimization of that empirical loss, which should correspond to fundamental characteristics of the data that are generalizable, as we show through the robust performance on out-of-distribution datasets.
> One potential scenario where a detrimental effect could appear is when the data exhibits distributional symmetry breaking, e.g., on the RotatedMNIST dataset. In that case, for instance, the rotation augmentations would lead the model to struggle to distinguish a 6 and 9. However, our results in Section 5.1 and Appendix A.4 show that there is an overall positive effect on the dataset that can clearly overcome that potential negative effect.

---

### Author Response · Authors · 2025-12-03
**Summary of contributions and main revisions**

We thank the new Area Chair for their additional effort for our submission under the updated review process, and we would also like to thank the reviewers for their insightful comments and suggestions, which have helped us improve the paper. Below, we provide a brief summary of our paper and the main revisions made after the reviews and discussion.

Equivariant machine learning methods are powerful for handling symmetry in geometric data, especially in scientific domains. However, they typically rely on the strong assumption that the relevant symmetries in the data are known exactly in advance, which significantly limits their practical applicability. To address this, we introduce a model-agnostic framework for learning approximate equivariances to a priori unknown symmetries. Our approach is based on a learnable data augmentation module that discovers the underlying data symmetries in an interpretable way, enabling the construction of approximately equivariant methods without requiring explicit prior knowledge of the symmetry group.

Specifically, our method discovers continuous symmetries through a reparameterization of Lie groups with their associated Lie algebras, and then samples elements of the learned group to augment the inputs before they are fed to a prediction model. To guide the discovery of the correct symmetry, we employ a multi-task loss that combines a standard empirical loss term (e.g., for prediction accuracy) with an additional term that penalizes equivariance error.

In the revised manuscript, we have:

- Extended our experimental analysis. We added new experiments in Appendix A.10 on a dataset without symmetry, and in Appendix A.11 on Quantum Field Theory amplitude prediction. These results further illustrate the versatility and broad applicability of our approach across different domains.
- Clarified our novel contributions and positioned them more clearly in the related work. Particularly, our method distinguishes itself through its end-to-end supervised training and provides interpretability for the learned symmetry.
- Further emphasized the ablations and robustness of our approach, in which we highlighted more in the paper the ablation results in Appendix A.9, demonstrating the robustness of our method to different choices of hyperparameters.
- Included a more detailed description of the symmetry parameterization. We expanded the explanation of how symmetries are parameterized in our framework, clarifying which groups can be represented and therefore discovered by our method.

We hope this summary of our work and the main applied revisions helps to provide a clear overview of our contributions that can complement our discussion with the reviewers.

We thank the Area Chair and reviewers once again for their time and effort, especially given the extraordinary circumstances of this review period.

---

### Meta-Review · Area_Chair_5ijU · 2025-12-23

**Summary:**

The paper introduces a method to discover symmetries by learning a distribution over a Lie matrix group, which can be sampled to augment a given input, and enables end-to-end training with a model that takes the augmented inputs.

Reviews were mixed. Reviewers praised the simplicity and interpretability of the method, extensive experiments and ablations. Several concerns were raised about its limitation to low-dimensional, well-behaved and connected Lie groups, efficiency, poor explanations, innacuracies, and missing references.

The rebuttal answered some questions but I don't it was enough to revert the two initial scores of 2 to an acceptance. Thus, I recommend rejection.

**Reviewer Concerns:**

Several reviewers raised concerns with the submission's limitation to low-dimensional, well-behaved and connected Lie groups, while some related work have no such constraints. While the rebuttal touched the subject and mentioned that the related work is addressing slightly different tasks, it could not really resolve this concern without significant modifications to the method.

A related concern was about the use of toy datasets in most experiments on which the choice of low-dimensional matrix-groups assumes a prior on the kinds of symmetries that can be learned. Such concern also cannot be addressed without significant effort and it's unclear whether the method could work on more complex scenarios.

Some concerns were resolved: discussion of efficiency, sensitivity analysis, and experiments on the Lorenz group.

**Reviewer Scores:**

X127, Sgvd were leaning accept and I don't think they would change their scores.

UrkV, VRqe had initial scores of 2 and both raised issues with the simplicity of the groups considered. I do not think they would revert to an accepting score after the rebuttal.

---

### Decision · Program_Chairs · 2026-01-26

Reject